

# Data worth analysis within a model-free data assimilation framework for soil moisture flow

Yakun Wang[1], Xiaolong Hu[2], Lijun Wang[2], Jinmin Li[2], Lin Lin[2], Kai Huang[3], Liangsheng Shi[2*]

[1]Key Laboratory of Agricultural Soil and Water Engineering of in Arid and Semiarid Areas, Ministry of Education, Northwest A & F University, Yangling, Shaanxi 712100, China

[2]State Key Laboratory of Water Resources and Hydropower Engineering Sciences, Wuhan University, Wuhan, Hubei 430072, China

[3]Guangxi Key Laboratory of Water Engineering Materials and Structures, Guangxi Institute of Water Resources Research, Nanning530023, China

*Correspondence to*: Liangsheng Shi (liangshs@whu.edu.cn)

**Abstract.** Conventional data-worth (DW) analysis for soil water problems depends on physical dynamic models. The widespread occurrence of model structural errors and the strong nonlinearity of soil water flow may lead to biased or wrong worth assessment. By introducing the nonparametric data-worth analysis (NP-DWA) framework coupled with ensemble Kalman filter (EnKF), this real-world case study attempts to assess the worth of potential soil moisture observations regarding the reconstruction of fully data-driven soil water flow models prior to data gathering. The DW of real-time soil moisture observations after Gaussian process training and Kalman update was quantified with three representative information metrics, including the trace, Shannon entropy difference, and relative entropy. The sequential NP-DWA framework was examined by a number of cases in terms of the variable of interest, spatial location, observation error, and prior data content. Our results indicated that the overall increasing trend of the DW from the sequential augmentation of additional observations was susceptible to interruptions by localized surges due to never-experienced atmospheric conditions (i.e., rainfall events) within the NP-DWA framework. Fortunately, this performance degradation can be effectively alleviated by enriching training scenarios or the appropriate amplification of observational noise under extreme meteorological conditions. Nevertheless, a substantial expansion of the prior data content may cause an unexpected increase in DW of future potential observations due to the possible introduction of ensuing observation noises. Hence, high-quality and representative "small" data may be a better choice than unfiltered "big" data. Compared with the observations in the surface layer with the strongest time-variability, the soil water content in the middle layer robustly exhibited remarkable

superiority in the construction of model-free soil moisture models. An alternative monitoring strategy with a larger data-worth was prone to a higher DW assessment accuracy within the proposed NP-DWA framework. We also demonstrated that the DW assessment performance was jointly determined by '3C', i.e., capacity of potential observation realizations to "capture" actual observations, correlation of

potential observations with the variables of interest, and choice of DW indicators. Direct mapping from regular meteorological data to soil water content within the NP-DWA mitigated the adverse effects of nonlinearity-related interference, which thus facilitated the identification of the soil moisture covariance matrix, especially the cross-covariance.

*Keywords*: Data worth; Nonparametric data assimilation; Soil moisture; Gaussian process


## 1 Introduction

As one of the few directly observable hydrological variables, soil water content (SWC) exhibits critical importance in optimal water resource management, irrigation and drainage schemes, fertilizer application, and crop production in agriculture (Liu et al., 2011; Akhtar et al., 2019; Dobriyal et al.,

2012; Gu et al., 2021). Various data assimilation (DA) approaches (Dunne and Entekhabi, 2005; Li and Ren, 2011; Reichle et al., 2008; Song et al., 2014) have been established to reconstruct the spatiotemporal dynamics of SWC from noisy or partial observations. The core of these traditional parametric filters is their reliance on repeated forward integrations of an explicitly known physical model of unsaturated flow, such as the HYDRUS (Šimůnek et al., 2006), Soil and Water Assessment

Tool (SWAT) (Van Dam and Feddes, 2000), and Ross models (Ross, 2003; Zha et al., 2013).

Currently, the ever-increasing availability of multi-source data from remote sensing (Montzka et al., 2011; Shi et al., 2011), ground-based measurements (Li et al., 2018; Shuwen et al., 2005; Yang et al., 2000), and numerical modeling has paved the way for the development of fully data-driven techniques within the DA framework. In particular, recent advances in machine learning-based DA schemes

(Brajard et al., 2020; Brajard et al., 2021; Yamanaka et al., 2019) offer exciting new opportunities for extracting patterns and insights of soil moisture dynamics from data (Ju et al., 2018; Li et al., 2020; Liu et al., 2020; Wang et al., 2021a). For instance, Kashif Gill et al. (2007) proposed a hybrid DA methodology that combined support vector machines and the ensemble Kalman filter (EnKF) for soil moisture dynamics. Li et al. (2020) compared the performance of a physical-based model with DA and

machine learning methods in terms of the simulation of soil water dynamics under synthetic and





real-world conditions. Wang et al. (2021a) and Wang et al. (2021b) further attempted to learn unknown relationships between SWC as well as its spatio-temporal gradients and highly accessible data via the Gaussian process (GP) regression.

Notwithstanding the success of these model-free DA schemes built on machine learning for

unsaturated flow, essential caveats and limitations have hampered their further adoption and impact. First, the amount of data required to infer nonlinear relationships in unsaturated flow problems may be overwhelming (Hughes, 1968), thus greatly increasing the data collection budget. Subsequently, addressing the abovementioned explosive data growth is also a challenging and time-demanding task requiring an extensive computational infrastructure. Second, the performance and quality of the

knowledge extracted by machine learning algorithms are highly dependent on the quality and suitability of data (García-Gil et al., 2019). Unfortunately, data gathering is rarely perfect, and data corruption often occurs (Wang et al., 2018). The identification of the multi-source SWC data quality or measurement error is not an easy task. This limitation instead can create extra uncertainties in DA systems (Kisekka et al., 2015) . Third, it is the diversity of scenarios contained in prior data rather than

its volume that is more decisive for the generalization ability of machine learning methods (Wang et al., 2020) . Direct data fusion without screening may instead induce accidental correlations in learning algorithms, thereby diminishing their generalization ability (García et al., 2016). To avoid the overloaded monitoring cost due to redundant data, it is essential to develop a framework to assess the worth of alternative sampling strategies prior to data collection.

Data worth, sometimes called data information content or data impact, of a design is often defined as its individual capability to reduce uncertainty associated with a prediction goal, or to maximize some related measure of data utility. Over the past decades, two main types of sophisticated DW analysis frameworks have been proposed to identify the most informative monitoring strategy in hydrology, namely, one type based on sensitivity analysis (Dausman et al., 2010; Fienen et al., 2010; Hill and

Tiedeman, 2006) and the other within a fully Bayesian framework (Dai et al., 2016; Neuman et al., 2012; Nowak et al., 2012). The former approaches are computationally fast, but these methods require model calibration and assume linear models (Finsterle, 2015). The latter methods are derived based on the law of the total possibility, without assumptions of the model and of the distributions of observations and model parameters. Nevertheless, both well-established frameworks are predicated on

the availability of the underlying physical models. For example, Man et al. (2016) evaluated the



expected value of alternative SWC sampling strategies with respect to the estimation of soil hydraulic parameters in the Hydrus-1D model, while Finsterle (2015) examined the worth of datasets potentially applicable for the calibration of geothermal reservoir models. Within such parametric data-worth analysis frameworks, however, the strong nonlinearity of soil water problems (De Lannoy et al., 2006;

Leube et al., 2012; Yeh et al., 1985) and the prevalence of model structural errors (Zhang et al., 2019) are highly likely to lead to biased data worth (Wang et al., 2018; Wang et al., 2020). Ultimately the reliability of optimal design of monitoring networks based on such evaluations is greatly compromised.

Fortunately, the superiority of data-driven algorithms in handling nonlinearities and structural errors in unsaturated flow has been well demonstrated in our previous studies (Wang et al., 2021a; Wang et

al., 2021b). With the explosive growth of big data, how to evaluate the worth of multi-source data in this new data mining approach is becoming a critical issue. Several recent works in the field of statistical learning have bloomed in identifying and removing irrelevant and redundant information from big data, such as feature selection (Chandrashekar and Sahin, 2014; Hall, 1999) and instance reduction (Al-Akhras et al., 2021; Olvera-López et al., 2010). To the best of our knowledge, few

studies have systematically evaluated the worth of future observations regarding the construction of fully data-driven models prior to data gathering. As a follow-up study of Wang et al. (2018) and Wang et al. (2020), one major contribution of this study is the first embedding of a purely data-driven model into the Bayesian data-worth analysis framework, referring to as the nonparametric data-worth analysis (NP-DWA). Similar to traditional DW analysis, the proposed NP-DWA consists of prior, posterior, and

preposterior stages (Dai et al., 2016). The preposterior analysis evaluates the anticipated worth of future observations regarding the construction of purely data-driven models, for which possible distributions are predicted in advance by conditioning on prior data.

There is a consensus in the field of statistics that "the highest accuracy results that an inductive learning system can achieve depend on the quality of data and on the appropriate selection of a learning

algorithm for the data" (Pechenizkiy et al., 2006). In other words, once the algorithm specified, the significance of data noise on learning accuracy as almost the only factor should not be overlooked. Considering the powerful ability of dealing with observational noises of the ensemble Kalman filter (EnKF) (Hamilton et al., 2017; Li et al., 2018), another innovation of this study is the introduction of EnKF into our NP-DWA framework. In conventional DW analysis, the worth of data is primarily

embodied in its ability to be utilized or calibrated to adjust physical parameters (Dai et al., 2016;



Finsterle, 2015; Man et al., 2016). In the proposed NP-DWA, nevertheless, future observations are first used to construct data-driven models in the forecast step and then sequentially assimilated with the Kalman update in the analysis step. Ultimately, its combined capacity to reduce system uncertainties in these two ways is defined as its worth. Furthermore, as a typical sequential DA scheme, the EnKF

facilitates the dynamic models as well as its hyperparameters to be updated in real time, so the data utility to modeling system can be detected instantaneously. Eventually, the sampling scheme can be dynamically adjusted to save the monitoring and analysis costs.

Most previous studies are based on synthetic cases, and data-worth analysis in the context of dynamically evolving soil moisture profiles was still poorly studied in a real-world case. For nonlinear

problems, nevertheless, the estimation variance and more sophisticated measures of data utility depend on the actual values of measurements, which are still unknown prior to collection (Leube et al., 2012). It will be more convincing to investigate the data worth regarding the reconstruction of fully data-driven models under real-world cases for unsaturated flow. With the aid of observed data retrieved from three typical stations with different climate regimes, we aim to shed light on the following

questions: (1) as opposed to the traditional way of utilizing data (to calibrate physical parameters), is the worth of observations capable of being accurately quantified by NP-DWA in this new, purely data-driven approach? (2) Given multiple prediction objectives, how does the DW (in the form of various indices) evolve under different hydrometeorological conditions in the determination of fully data-driven soil moisture dynamics? (3) How does the proposed NP-DWA respond to the presence of

multiple levels of data noise? It is strived that this study can provide guidance in the design of future monitoring strategies within the fully data-driven soil water flow models for real-world problems.

The remainder of this paper is organized as follows: Sect. 2 first summarizes the experimental data and methods. Thereinto, the principles of Bayesian DW analysis, nonparametric DA, and the hybrid framework are detailed. Sect. 3 presents the results, and a discussion is contained in Sect. 4. Finally,

conclusions are outlined in Sect. 5.

## 2 Methodology

In Wang et al. (2021a), a nonparametric sequential data assimilation scheme (Kalman-GP) has been proposed based on the filtering equations of EnKF and data-driven modeling with GP. On top of that, this paper further develops a nonparametric data-worth analysis framework to assess the potential

worth of future observations in the reconstruction of dynamical soil water flow models prior to data





collection. Considering that the Kalman-GP have been described in detail in Wang et al. (2021a), only a brief introduction to it is presented here.

### 2.1 Construction of GP Dynamic Models

We aim to determine a Gaussian stochastic process to approximate the relationship between input data $\boldsymbol{x}$ and state variable of interest $y$. It should be noted that input $\boldsymbol{x}$ may consist of any relevant information in addition to the time and location of quantity of interest. As defined in Rasmussen (2003) and Williams and Rasmussen (2006), a GP $G(\boldsymbol{x})$ can be fully specified by a mean function $\mu(\boldsymbol{x})$ and covariance function $k(\boldsymbol{x}, \boldsymbol{x}')$, i.e., $G(\boldsymbol{x}) \sim N(\mu(\boldsymbol{x}), k(\boldsymbol{x}, \boldsymbol{x}'))$. In this study, a linear mean function and an anisotropic squared exponent covariance function are specified (Zhang et al., 2019) as:

$$\mu(\boldsymbol{x}) = \boldsymbol{\beta}^{\mathrm{T}} \boldsymbol{x} \tag{1}$$

$$k(\boldsymbol{x}, \boldsymbol{x}') = \sigma^2 exp\left[-\sum_{l=1}^{d} \frac{(x_l - x_l')^2}{\tau_l^2}\right] \tag{2}$$

where $\boldsymbol{\beta}$ is vector containing $d$ linear coefficients, i.e., $\boldsymbol{\beta} = \{\beta_1, \beta_2, \dots, \beta_d\}$; $d$ is the dimension of $\boldsymbol{x}$; $\sigma^2$ controls the marginal variance in the output; and $\tau_1, \tau_2, \dots, \tau_d$ determine the dependence strength in each of the component directions of $\boldsymbol{x}$.

Next, let $\boldsymbol{X} = \{\boldsymbol{x}^i\}_{i=1}^{N}$ denote the input of $N$ training datasets, while the corresponding output can be represented as $\boldsymbol{y} = (y^1, y^2, \dots, y^N)^{\mathrm{T}}$. Then, the hyperparameters of the GP, $\boldsymbol{\emptyset} = \{\boldsymbol{\beta}, \sigma^2, \boldsymbol{\tau}\}$, can be inferred from the training datasets $\{\boldsymbol{X}, \boldsymbol{y}\}$ via log marginal likelihood maximization:

$$L = \log p(\boldsymbol{y}|\boldsymbol{X}, \boldsymbol{\emptyset}) = -\frac{1}{2}(\boldsymbol{y} - \boldsymbol{\mu})^{\mathrm{T}} \Sigma^{-1}(\boldsymbol{y} - \boldsymbol{\mu}) - \frac{1}{2}\log|\Sigma| - \frac{n}{2}\log 2\pi \tag{3}$$

where $\boldsymbol{\mu}$ denotes the prior mean vector, and $\boldsymbol{\Sigma}$ denotes the covariance matrix whose elements in the $i$th row and $j$th column constitute $\Sigma_{ij} = k(\boldsymbol{x}_i, \boldsymbol{x}_j)$. The GPML MATLAB toolbox (version 4.2), as documented in Williams and Rasmussen (2006), was adopted for GP inference in this study (http://www.gaussianprocess.org/gpml/code/matlab/doc/).

Finally, the posterior mean $\boldsymbol{y}^*$ can be predicted for any new input $\boldsymbol{X}^*$ as:

$$\boldsymbol{y}^* = \boldsymbol{\mu}^* + \boldsymbol{\Sigma}^{*\mathrm{T}} \boldsymbol{\Sigma}^{-1}(\boldsymbol{y} - \boldsymbol{\mu}) \tag{4}$$

where $\boldsymbol{\mu}^*$ denotes the prior mean of $\mu(\boldsymbol{X}^*)$ and $\boldsymbol{\Sigma}^*$ is calculated as $\Sigma_i^* = k(\boldsymbol{x}_i, \boldsymbol{X}_j^*)$.

### 2.2 The Kalman Update in Nonparametric Data Assimilation Scheme

Similar to the conventional EnKF method (Evensen, 2003), the model-free DA strategy also comprises forecast and analysis steps. At the forecast step at $t=k$, $N_1$ GP dynamic models are





constructed    in    the    light    of    $\left\{(\boldsymbol{X}_{1:k-1}, \boldsymbol{y}_{1:k-1})_1, \dots, (\boldsymbol{X}_{1:k-1}, \boldsymbol{y}_{1:k-1})_i, \dots, (\boldsymbol{X}_{1:k-1}, \boldsymbol{y}_{1:k-1})_{N_1}\right\}$

independently via Eqs. 1-3. Here, $(\boldsymbol{X}_{1:k-1}, \boldsymbol{y}_{1:k-1})$ represents all available data points from $t$=1 to ($k$-1)

and $i$ is the ensemble member index. Hence an ensemble of $\boldsymbol{y}^*$ at the current time step, $\boldsymbol{Y}_k^f$, can be

forecasted via Eq. 4, which can be expressed as:

$$\boldsymbol{Y}_k^f = \left[\boldsymbol{y}_{k,1}^*, \boldsymbol{y}_{k,2}^*, \dots, \boldsymbol{y}_{k,i}^*, \dots, \boldsymbol{y}_{k,N_1}^*\right]^{\mathrm{T}} \tag{5}$$

where superscripts $f$ refers to forecast; $\boldsymbol{y}_{k,i}^*$ represents the forecasted state vector of interest for the $i$th

GP model that was constructed at $t$=$k$.

In the analysis step of the EnKF, the resultant forecasted state vector at $t$=$k$, $\boldsymbol{Y}_k^f$, is updated via the

assimilation of current observation data, $\boldsymbol{d}_k^{obs}$:

$$\boldsymbol{Y}_k^a = \boldsymbol{Y}_k^f + \boldsymbol{K}_k(\boldsymbol{d}_k^{obs} - \boldsymbol{H}\boldsymbol{Y}_k^f) \tag{6}$$

where $\boldsymbol{Y}_k^a$ denotes the posterior estimates for the ensemble of state vectors conditional on the observed

data at $t$=$k$; superscript $a$ indicates analysis; and $\boldsymbol{H}$ is the observation operator, which represents the

relationship between the state and observation vectors.

The Kalman gain at $t$=$k$, $\boldsymbol{K}_k$, can be defined as:

$$\boldsymbol{K}_k = \boldsymbol{C}_k^f \boldsymbol{H}^T (\boldsymbol{H}\boldsymbol{C}_k^f \boldsymbol{H}^T + \boldsymbol{R}_k)^{-1} \tag{7}$$

where $\boldsymbol{R}_k$ is the error covariance matrix of the observations; and $\boldsymbol{C}_k^f$ is the covariance matrix of the

state vector at $t$=$k$, which can be approximated as:

$$\boldsymbol{C}_k^f \approx \frac{1}{N_e - 1} \sum_{i=1}^{N_1} \left\{ \left[\boldsymbol{y}_{k,i}^f - \langle\boldsymbol{Y}_k^f\rangle\right]\left[\boldsymbol{y}_{k,i}^f - \langle\boldsymbol{Y}_k^f\rangle\right]^{\mathrm{T}} \right\} \tag{8}$$

where $\boldsymbol{y}_{k,i}^f$ is equivalent to $\boldsymbol{y}_{k,i}^*$ and $\langle\boldsymbol{Y}_{k,i}^f\rangle$ denotes the ensemble mean of $\boldsymbol{Y}_k^f$.

**2.3 Nonparametric Data-worth Analysis Framework**

Following the methodologies of Neuman et al. (2012) and Dai et al. (2016), data-worth analysis of

future monitoring networks within the aforementioned NP-DWA framework also consists of three

stages. The whole workflow of the NP-DWA framework is depicted in Fig. 1.

**2.3.1 Prior Stage**

At the prior stage ($0 < t \le T_p$), the integration of GP dynamic models and EnKF with an ensemble

size of $N_1$ is implemented to sequentially train and assimilate the prior data via Eqs. 1–8. Here, all

available prior datasets from $t$=0 to $t$=$T_p$ are denoted as $\boldsymbol{A} = \boldsymbol{y}_{1:T_p} = \boldsymbol{d}_{1:T_p}^{obs}$, while the corresponding

GP input is denoted as $\boldsymbol{X}_{1:T_p}$. Then, a set of $N_e$ hypothetical observations can be generated, denoted as





$\boldsymbol{B}_{k,i} = \boldsymbol{H}_k \boldsymbol{y}_{k,i}^f$ $(k = T_p + 1, T_p + 2, \dots, T_t; \ i = 1,2,\dots,N_1)$, via Eq. 4. $T_t$ is the total simulation time.

Moreover, prior prediction statistics (mean and covariance) of posterior vector $\boldsymbol{Y}_k$, i.e., $E(\boldsymbol{Y}|\boldsymbol{A})$ and

$Cov(\boldsymbol{Y}|\boldsymbol{A})$, can be yielded conditional on $\{\boldsymbol{A}\}$, which can be denoted as $\boldsymbol{E}_1$ and $\boldsymbol{C}_1$, respectively, for

the sake of simplicity.

***Prior Stage:***
➢ Train $N_1$ GP dynamic models in the light of $\left\{\boldsymbol{X}_{1:T_p}, \boldsymbol{A} = \boldsymbol{y}_{1:T_p} = \boldsymbol{d}_{1:T_p}^{obs}\right\}$ 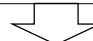
➢ Assimilate prior available data $\boldsymbol{A}$ to generate additional potential observations $\boldsymbol{B}$ by the EnKF
➢ Calculate the prior prediction statistics conditional on $\{\boldsymbol{A}\}$, i.e., $E(\boldsymbol{Y}|\boldsymbol{A})$ and $Cov(\boldsymbol{Y}|\boldsymbol{A})$

***Preposterior Stage:***
➢ Train GP dynamic models with $\{\boldsymbol{A}, \boldsymbol{B}\}$ and sequentially assimilate them to obtain the resultant statistics including $E_{\boldsymbol{B}|\boldsymbol{A}}E(\boldsymbol{Y}|\boldsymbol{A}, \boldsymbol{B})$ and $E_{\boldsymbol{B}|\boldsymbol{A}}Cov(\boldsymbol{Y}|\boldsymbol{A}, \boldsymbol{B})$ 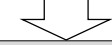
➢ Calculate the expected data-worth in the form of $T_r$, $SED$, and $RE$

***Posterior Stage:***
➢ Train GP dynamic models with $\boldsymbol{A}$ and additionally available data $\boldsymbol{B}'$ and sequentially assimilate them to obtain the posterior statistics including $E(\boldsymbol{Y}|\boldsymbol{A}, \boldsymbol{B}')$ and $Cov(\boldsymbol{Y}|\boldsymbol{A}, \boldsymbol{B}')$
➢ Calculate the reference data-worth in the form of $T_r$, $SED$, and $RE$

**Figure 1.** The workflow of nonparametric data-worth analysis framework coupled with Ensemble Kalman Filter

(EnKF)

### 2.3.2 Preposterior Stage

At the preposterior stage $(T_p + 1 < t \le T_t)$, for each possible data $\boldsymbol{B}_{k,i}$ at $t=k$, $N_2$ realizations

satisfying a Gaussian distribution are further generated. The ensemble mean is the value of $\boldsymbol{B}_{k,i}$, while

the variance is the measurement error. Since this method is recursive, the time index $k$ is omitted in the

following equations. Then, the integration of GP models and EnKF is again implemented through a set

of $N_2$ Monte Carlo realizations for each of the $N_1$ hypothetical observations. This allows us to

calculate prediction statistics of the posterior state vector $\boldsymbol{Y}_{ij}$ $(i = 1,2,\dots,N_1; \ j = 1,2,\dots,N_2)$, i.e.,

$E(\boldsymbol{Y}_i|\boldsymbol{B}_i)$ and $Cov(\boldsymbol{Y}_i|\boldsymbol{B}_i)$, conditional on $\{\boldsymbol{A}, \boldsymbol{B}_i\}$. Finally, quantities $E_{\boldsymbol{B}|\boldsymbol{A}}E(\boldsymbol{Y}|\boldsymbol{A}, \boldsymbol{B})$,

$E_{\boldsymbol{B}|\boldsymbol{A}}Cov(\boldsymbol{Y}|\boldsymbol{A}, \boldsymbol{B})$, and $Cov_{\boldsymbol{B}|\boldsymbol{A}}E(\boldsymbol{Y}|\boldsymbol{A}, \boldsymbol{B})$ can be yielded by averaging over the collection of $N_1 \times N_2$

realizations. It should be emphasized that $E_{\boldsymbol{B}|\boldsymbol{A}}E(\boldsymbol{Y}|\boldsymbol{A}, \boldsymbol{B})$ and $E_{\boldsymbol{B}|\boldsymbol{A}}Cov(\boldsymbol{Y}|\boldsymbol{A}, \boldsymbol{B})$ represent the

preposterior prediction mean and uncertainty after the addition of future possible data $\boldsymbol{B}$, which can be





denoted as $\boldsymbol{E}_2$ and $\boldsymbol{C}_2$, respectively.

To quantify the expected data-worth of potential measurements, three commonly considered

information metrics, including the trace ($T_r$), Shannon entropy difference ($SED$), and relative entropy

($RE$), are introduced in this study. $T_r$ and $SED$ offer scalar indices to measure the decrease in variance

and covariance, respectively, while the $RE$ comprehensively quantifies both mean and covariance

effects.

(1) Trace

As a scalar indicator (Dai et al., 2016), $T_r$ quantifies the DW in terms of variance reduction as

follows:

$$T_r = T_r(\boldsymbol{C}_1) - T_r(\boldsymbol{C}_2) \tag{9}$$

where $T_r(*)$ denotes the trace (sum of the diagonal entries) of a matrix.

(2) Shannon entropy difference

According to Shannon (1949), the Shannon entropy ($SE$) of *PDF* $p(x)$ can be defined as:

$$SE(p) = -\int p(x) \ln p(x) \, dx, x \in R \tag{10}$$

The $SED$ between the prior and preposterior PDFs can also be considered to quantify the information

content extracted from additional observations. Assuming that these two PDFs are both Gaussian in the

EnKF model, the $SED$ can be expressed in terms of covariance reduction (Xu, 2007) as:

$$SED = \frac{\ln \det(\boldsymbol{C}_1)}{2} - \frac{\ln \det(\boldsymbol{C}_2)}{2} = \frac{\ln \det(\boldsymbol{C}_1\boldsymbol{C}_2^{-1})}{2} \tag{11}$$

where $\det(*)$ is the determinant of a matrix.

(3) Relative entropy

Similar to the $SED$, the $RE$ also provides a measure of the information content of the preposterior

PDF with respect to the prior PDF. In addition to uncertainty reduction, the influence of future data on

the mean behavior of PDFs is considered (Singh et al., 2013; Zhang et al., 2015). Considering that the

prior and preposterior PDFs are $n$-dimensional Gaussian functions, the $RE$ can be defined as:

$$RE = \frac{1}{2}(\boldsymbol{E}_2 - \boldsymbol{E}_1)^T \boldsymbol{C}_1^{-1}(\boldsymbol{E}_2 - \boldsymbol{E}_1) + \frac{1}{2}[\ln \det (\boldsymbol{C}_1\boldsymbol{C}_2^{-1}) + T_r(\boldsymbol{C}_2\boldsymbol{C}_1^{-1}) - n] \tag{12}$$

Finally, the expected DW of $\boldsymbol{B}_k$ can be estimated in the form of the above three indices prior to data

gathering. Similar procedures are repeated until the final time $t=T_t$ is reached.

### 2.3.3 Posterior Stage

At the posterior stage ($T_p + 1 < t \leq T_t$), the available actual dataset $\boldsymbol{B}'$ is incorporated into the GP



training datasets and assimilated in a sequential manner. The actual mean and covariance of posterior state vector $Y$, i.e., $E(Y|A, B')$ and $Cov(Y|A, B')$, respectively, are obtained conditional on $\{A, B'\}$.

The reference data-worth in the form of the various indices can be calculated via Eqs. 9–12, where $E_2$ and $C_2$ are replaced with $E(Y|A, B')$ and $Cov(Y|A, B')$, respectively.

## 3 Description of experimental data and model setup

### 3.1 Data Sources and Site Description

Three typical sites, including Falkenberg (52.1669 N, 14.1241 E), Cape_Charles_5_ENE (37.2907 N,

75.9270 W, hereafter referred to as Cape), and DAHRA (15.4035 N, 15.4320 W) were selected from the International Soil Moisture Network (ISMN, https://ismn.geo.tuwien.ac.at/en/) to evaluate the performance of the proposed NP-DWA framework under different soil types and climatic regimes. According to the dominant fraction of clay, silt, and sand for two layers (topsoil: 0.0–0.3 m, subsoil: 0.3–1.0 m) provided by ISMN, we use the USDA soil texture classification and classified the soil at

three sites. The soil at Falkenberg is sandy loam, and the DAHRA soil is loamy sand. The topsoil and subsoil at Cape are clay loam and loamy clay, respectively. At these three sites, the in situ volumetric SWC was operationally measured with TRIME-EZ (IMKO), Stevens Hydraprobe II Sdi-12 (Stevens Water Inc.), and ThetaProbe ML2X (Delta-T Devices) instruments, respectively. The measurement depths were (1) 0.08, 0.15, 0.30, 0.45, 0.60, and 0.90 m at the Falkenberg site, (2) 0.05, 0.10, 0.20, 0.50,

and 1.00 m at the Cape site, and (3) 0.05, 0.10, 0.50, and 1.00 m at the DAHRA site. The measurement error was assumed to be 0.02 cm$^3$/cm$^3$ unless otherwise specified.

Apart from soil water measurements at different depths, the daily precipitation and air temperature at the height of 2 meters were obtained from the ISMN. At each site, 200-day time series (from January 15 and August 2 in 2005 at the Falkenberg site, from April 24 to November 9 in 2004 at the Cape site,

and from April 9 to October 25 in 2011 at the DAHRA site) were collected in this study, as shown in Fig. 2. Having a continental climate, the Falkenberg receives frequent but less intense precipitation during the simulation period. The Cape has a humid subtropical climate with the highest rates rainfall among the three sites, and there were a few rainstorm events during the study period (e.g., up to 150 mm/d on September 8, 2014). The region of DAHRA has a tropical climate with well-defined dry and

humid seasons. The early stage of the simulation is in its dry season, with little to no rainfall. The late stage is in its humid season when frequent but less intense rainfall events occurs and the daily average air temperature is about 30 ◦C.

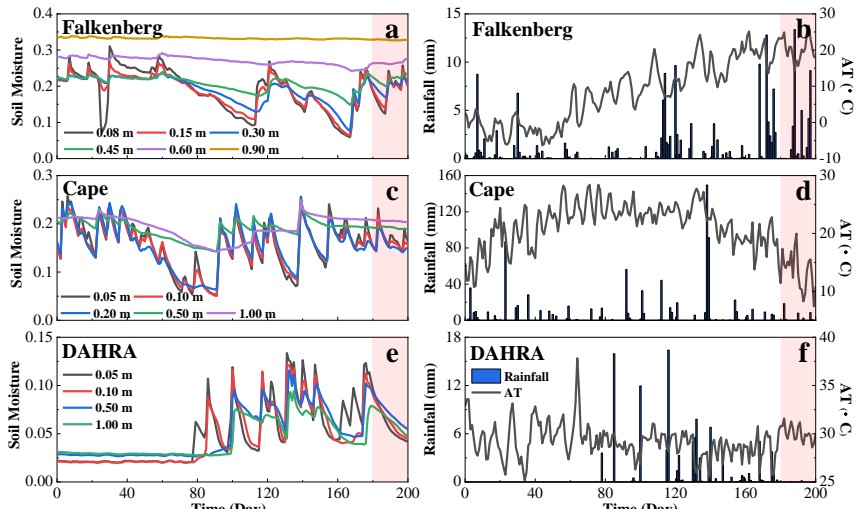

**Figure 2.** The temporal evolutions of soil moisture at various depths, daily rainfall, and mean daily air temperature

(AT) at 2 meters height at Falkenberg, Cape, and DAHRA, respectively. Note that the red area indicates the

preposterior or posterior stage

### 3.2 Model Simulation Setup and Case Design

The key parameters of this study are summarized in Table 1. Each site is represented by a

one-dimensional soil column with a height of 1 m, which is discretized into 2cm grids with local

refinement of 1-cm monitoring depth intervals, i.e., $z$=0.15 m and 0.45 m at the Falkenberg site and

$z$=0.05 m at the Cape and DAHRA sites. At each time step, $N_1 = 50$ GP-based dynamic models of

unsaturated flow are constructed. The GP model input $x$ includes the observation time, depth, daily

precipitation, and air temperature, while the output is the corresponding soil moisture. The state vector

$y$ comprises the soil moisture for all nodes at each site, and the trained and assimilated observations

$d^{obs}$ refer to the available soil moisture at all observed depths (as described in Sect. 3.1).

We illustrate our approach based on a set of real-world test cases, as listed in Table 2. The

performance of the three indices, namely, $T_r$, $SED$, and $RE$, in data-worth quantification are compared

at all three sites. In this study, the worth of potential observations regarding the retrieval of three

quantities of interest (QoI), including $\theta_{1.00}^{ave}$, $\theta_{0.60}^{ave}$, and $\theta_{0.30}^{ave}$, is evaluated. Here, $\theta_{1.00}^{ave}$, $\theta_{0.60}^{ave}$, and $\theta_{0.30}^{ave}$

represent the average soil moisture in the top 1.00 m, 0.60 m, and 0.30 m, respectively. A comparison

among cases TC1, TC2, and TC3, is designed to investigate the data-worth response of surface ($\theta_S$),





middle ($\theta_M$), and deep ($\theta_D$) SWC regarding the above different prediction objectives. The prior datasets

entering these cases comprise of SWC at various depths, daily precipitation, and air temperature over

the first 80 days, as shown in the gray areas of Fig. 2. The subsequent 20-day data (red areas in Fig. 2)

are augmented as additional data for reference DW assessment in the posterior stage.

**Table 1.** The summary of key parameters

| Parameter | Value |
|---|---|
| Description of soil column | |
| Soil column height [m] | 1.00 |
| No. of Nodes | 53 (Falkenberg)/52 (Cape&DAHRA) |
| Number of realizations | |
| $N_1$ | 50 |
| $N_2$ | 50 |
| Prior values of GP hyperparameters | |
| $\tau_1,\ \tau_2,\ \tau_3,\ \tau_4$ | 1 |
| $\sigma^2$ | 0.5 |
| $\beta_1,\ \beta_2,\ \beta_3,\ \beta_4$ | 0 |

**Table 2.** The summary of designed test cases and main characteristics

| Case Name | | Potential Observation | Observation Error | Prior Data (d) | Variable of Interest |
|---|---|---|---|---|---|
| TC1 | TC1-1 | $\theta_S$ | $0.02^2$ | 80 | $\theta^{ave}_{1.00}$ |
| | TC1-2 | $\theta_M$ | $0.02^2$ | 80 | $\theta^{ave}_{1.00}$ |
| | TC1-3 | $\theta_D$ | $0.02^2$ | 80 | $\theta^{ave}_{1.00}$ |
| TC2 | TC2-1 | $\theta_S$ | $0.02^2$ | 80 | $\theta^{ave}_{0.60}$ |
| | TC2-2 | $\theta_M$ | $0.02^2$ | 80 | $\theta^{ave}_{0.60}$ |
| | TC2-3 | $\theta_D$ | $0.02^2$ | 80 | $\theta^{ave}_{0.60}$ |
| TC3 | TC3-1 | $\theta_S$ | $0.02^2$ | 80 | $\theta^{ave}_{0.30}$ |
| | TC3-2 | $\theta_M$ | $0.02^2$ | 80 | $\theta^{ave}_{0.30}$ |
| | TC3-3 | $\theta_D$ | $0.02^2$ | 80 | $\theta^{ave}_{0.30}$ |
| TC4 | | $\theta_S$ | $0.01^2$ | 80 | $\theta^{ave}_{1.00}$ |
| TC5 | | $\theta_S$ | $0.04^2$ | 80 | $\theta^{ave}_{1.00}$ |
| TC6 | | $\theta_S$ | $0.02^2$ | 40 | $\theta^{ave}_{1.00}$ |
| TC7 | | $\theta_S$ | $0.02^2$ | 180 | $\theta^{ave}_{1.00}$ |
| TC8 | | $\theta_S,\ \theta_M$ | $0.02^2$ | 80 | $\theta^{ave}_{1.00}$ |
| TC9 | | $\theta_S,\ \theta_M,\ \theta_D$ | $0.02^2$ | 80 | $\theta^{ave}_{1.00}$ |





**Notes:** $\theta_S$, $\theta_M$, and $\theta_D$ refer to soil moisture in the surface, middle, and deep layers, respectively; $\theta^{ave}_{1.00}$, $\theta^{ave}_{0.60}$, and $\theta^{ave}_{0.30}$ refer to average soil moisture in the top 1.00 m, 0.60 m, and 0.30 m, respectively.

As stated in Pechenizkiy et al. (2006) and Zhu and Wu (2004), the maximum accuracy of statistical learning algorithms mainly depends on the quality of training data, in addition to the inherent bias in the algorithm itself. In other words, the magnitude and accuracy of the expected worth of driving data in machine learning-based DA may be closely related to the noise level. Thus, two additional test cases (TC4 and TC5) are considered to evaluate the performance of the proposed NP-DWA framework under different measurement errors. The soil moisture measurement error variance values of $0.01^2$ and $0.04^2$ are assumed in TC4 and TC5, respectively, to be compared to a value of $0.02^2$ in TC1-1.

Moreover, test cases TC6 and TC7 differ from test case TC1-1. These test cases are designed to investigate the influence of the prior data content on data-worth analysis, which facilitates the determination of the required prior information content to ensure the accuracy of data-worth assessment. The 80-day prior data in test case TC1-1 are reduced backward in time to 40 days in test case TC6 and augmented forward to 180 days in test case TC7. In addition, test cases TC1-1, TC8, and TC9 consider the composite DW of different combinations of monitoring schemes. The comprehensive contributions of the surface SWC jointly with the middle and/or deep ones are compared with its individual contribution.

**3.3 Evaluation setup**

To compare the relative differences in data-worth estimation accuracy under the various test scenarios, the mean absolute percentage error (MAPE) between the expected and reference data-worth in the form of $T_r$, *SED*, and *RE* is defined as:

$$MAPE = \frac{1}{T_t - T_p} \sum_{k=T_p+1}^{T_t} \left| \frac{DW_k^{Expect} - DW_k^{Refer}}{DW_k^{Refer}} \right| \tag{13}$$

where $DW_k^{Expect}$ and $DW_k^{Refer}$ denote the expected and reference DW values, respectively, at time step $t=k$.

**4 Results and discussions**

**4.1 Optimal Monitoring Location for the Multiple Predictive Objectives (TC1/TC2/TC3)**

Fig. 3 shows the probability distributions of the generated potential observation realizations as well as their ensemble mean and the corresponding actual observations of the surface ($\theta_S$), middle ($\theta_M$), and





deep ($\theta_D$) soil moisture at three sites. Only the results on the 81st, 90th, and 99th days are presented here. Overall, the $N_1 = 50$ potential realizations could "capture" the actual SWC observations with acceptable accuracy. Specifically, the forecasted middle SWC exhibited a considerably more robust

'capturing' performance with sustained better proximity of potential and actual $\theta_M$ throughout the simulation period. This occurred especially pronounced at the Cape site. For example, both surface and deep layers at Cape may be at risk of poor fit of potential observations to measurements (Fig. 3d and Fig. 3w), while the generated middle SWC is always fairly well approximated to the corresponding actual values in Fig. 3(m-o).

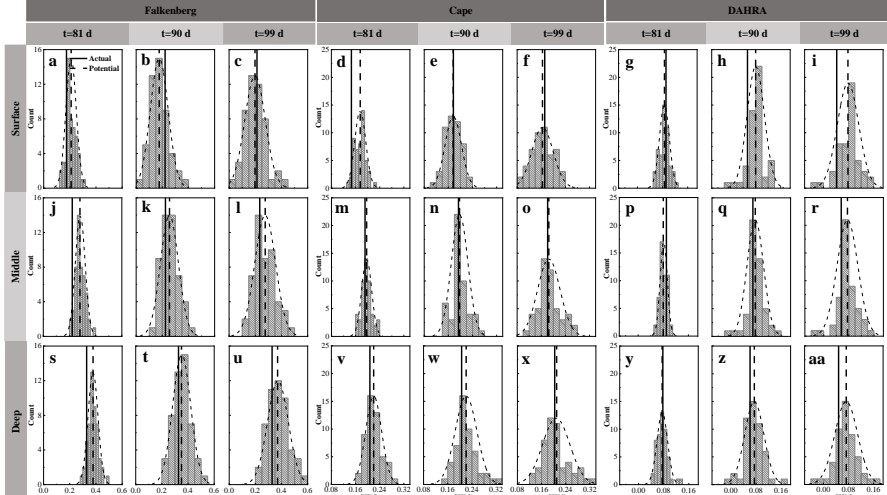


**Figure 3.** The probability distributions (dotted curved line) of potential observation realizations as well as their mean (dotted vertical line) and the corresponding actual soil water content (SWC) observation (solid line) in the surface, middle, and deep layers on the 81st, 90th, and 99th day at Falkenberg, Cape, and DAHRA, respectively

Based on the above potential observations, their expected data-worth regarding the retrieval of $\theta_{1.00}^{ave}$, $\theta_{0.60}^{ave}$, and $\theta_{0.30}^{ave}$ can be quantified in the form of $T_r$, $SED$, and $RE$, as depicted in Fig. 4. Meanwhile, for ease of analysis, Fig. 5 compares the covariance matrixes of entire soil moisture profile in the prior stage, preposterior stage, and posterior stage. Only the results from the 81st day to 90th day at Falkenberg are revealed here. It can be observed that despite an overall increasing trend over time, the

values of expected DW were prone to local spikes due to changes in the atmospheric boundary conditions such as rainfall. First of all, this general trend of increase should be attributed to the

sequential augmentation of potential observations based on existing prior data, resulting in the

cumulative values of DW over time. However, abrupt changes in external forcing, such as

unexperienced rainfall events on the 88[th] day at the Falkenberg, could trigger temporal extrapolation of

statistical learning (Li et al., 2020; Minns and Hall, 1996; Xu and Valocchi, 2015), which in turn led to

a surge in prior predictive uncertainty, i.e., $\boldsymbol{C_1} = Cov(\boldsymbol{Y}|\boldsymbol{A})$ (the 1[st] column of Fig. 5). Fortunately,

joint GP training and sequential assimilation of real-time potential observations can effectively lower

the risk of such irrational extrapolation (Wang et al., 2021a; Wang et al., 2021b), allowing these

temporal mutations to be substantially attenuated at the preposterior stage [i.e., $C_2 = E_{B|A}Cov(Y|A,B)$]

(the 2[nd]-4[th] column of Fig. 5). This uncertainty reduction brought about by the fusion of additional data

became significantly larger when external forcing encountered mutations, which ultimately led to the

localized surge in DW during rainfall events.

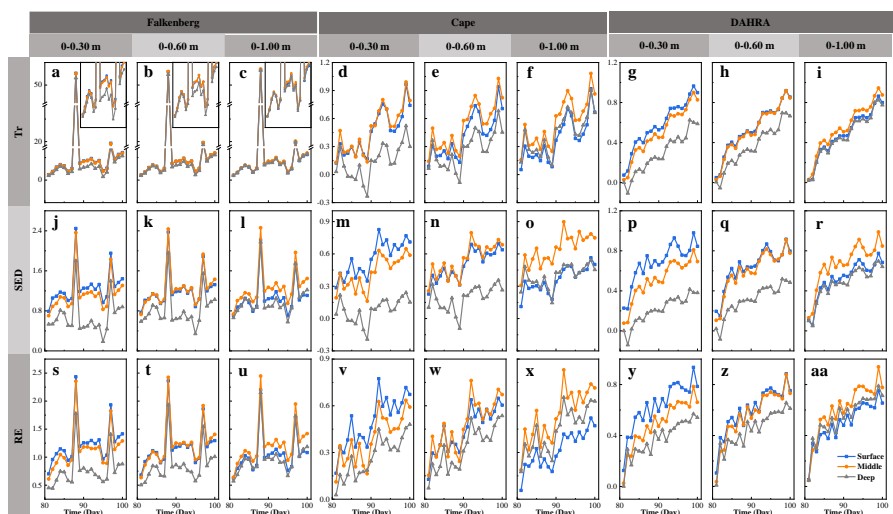

**Figure 4.** The expected data-worth of potential soil moisture observations in the surface, middle, and deep layers

in the form of trace ($T_r$), Shannon entropy difference (*SED*), and relative entropy (*RE*), respectively, regarding the

retrieval of average soil moisture in the top 0.30 m, 0.60 m, and 1.00 m at three sites

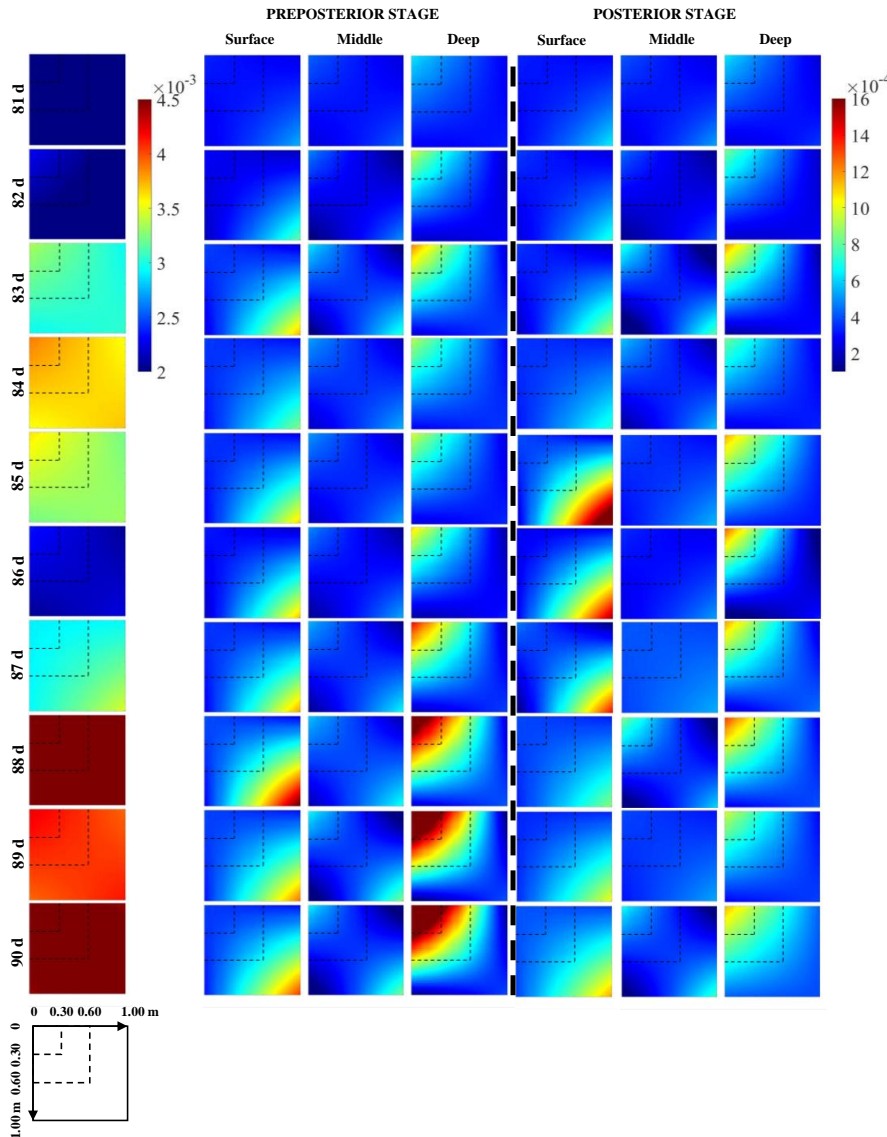

**Figure 5.** The covariance matrixes of soil moisture profiles from the 81$^{st}$ to 90$^{th}$ day at Falkenberg before (column 1) and after potential (columns 2-4) and corresponding actual (columns 5-7) soil moisture observations in the surface, middle, and deep layers were fused, respectively

Moreover, Fig. 4 also suggests that the optimal observation depth shifted as the prediction target changed. As expected, the surface SWC $\theta_S$ produced higher $T_r$, $SED$, and $RE$ values regarding the estimation of $\theta_{0.30}^{ave}$. As the depth range of the average SWC to be estimated was extended downward,



the data-worth advantages of $\theta_M$ and $\theta_D$ began to emerge. Surprisingly, the potential middle SWC still exhibited a considerably higher superiority even in $\theta_{1.00}^{ave}$ estimation. In other words, the soil moisture in the middle layer has the most robust advantage in data-worth. This may be due to the fact that the integration of surface or deep SWC only reduced the uncertainty within the corresponding depth ranges (the 2$^{nd}$ and 4$^{th}$ column of Fig. 5), whereas the augmentation of $\theta_M$ significantly decrease

the covariance matrixes of the entire SWC profiles (the 3$^{rd}$ column of Fig. 5). This selection result of the optimal monitoring location seemingly contradicts previous findings within the traditional parametric DW analysis where the surface observations with the largest temporal variation always produced the greatest data worth, as reported in Dai et al. (2016) and Wang et al. (2018). This discrepancy is likely to depend on the different mechanisms that characterize soil moisture dynamics in

the vertical direction between the two approaches. The traditional parametric unsaturated flow model follows the law of mass conservation-based physical governing equations (i.e., the Richardson–Richards equation, (Richards, 1931; Richardson, 1922)). The strongest time-varying nature of surface SWC was conducive to effective updating of the physical parameters in EnKF, eventually generating the maximum data-worth (Wang et al., 2018). However, the spatial prediction performance

of data-driven methods substantially hinged on the similarity of data between different depths. Theoretically, there occurs an inherent delayed response of soil moisture profiles to rainfall events, which has been well-documented experimentally (Wierenga et al., 1986; Bresler et al., 1971; Vauclin et al., 1979). This causes the temporal changes in surface and deep SWC to be naturally asynchronous, thus rendering their representativeness in characterizing the whole soil moisture profile somewhat

limited. Ultimately, the complete reliance on statistical and information-theoretic measures allowed the most representative middle SWC to establish the most robust superiority in DW.

It can also be seen from Fig.4 that when using different information indices (i.e., $T_r$, $SED$, and $RE$) to quantify the data-worth, the optimal observation location selected is identical, regardless of soil textures and climatic regimes. This conclusion is generally in line with Wang et al. (2018) and Man et

al. (2016). Furthermore, to quantify the data-worth assessment accuracy, Fig. 6 depicts the MAPE between the expected and reference data-worth in the form of $T_r$, $SED$, and $RE$ of alternative monitoring schemes at the different depths. It can be observed that the surface SWC yielded the smallest MAPE when retrieving $\theta_{0.30}^{ave}$, regardless of the metric type. During the estimation of $\theta_{0.60}^{ave}$ and $\theta_{1.00}^{ave}$, nevertheless, the expected data-worth of $\theta_M$ more accurately and robustly approached the





reference counterparts with overall smaller MAPEs. We recall that this ranking of DW estimation

accuracy was exactly in line with the ranking of the magnitude of their expected DW in Fig. 4. To be

specific, a comparison of Fig. 4 and Fig. 6 reveals that potential observations with a larger expected

DW are prone to a higher DW estimation accuracy due to its more robust ability of "imitating" the

actual observations (Fig. 3).

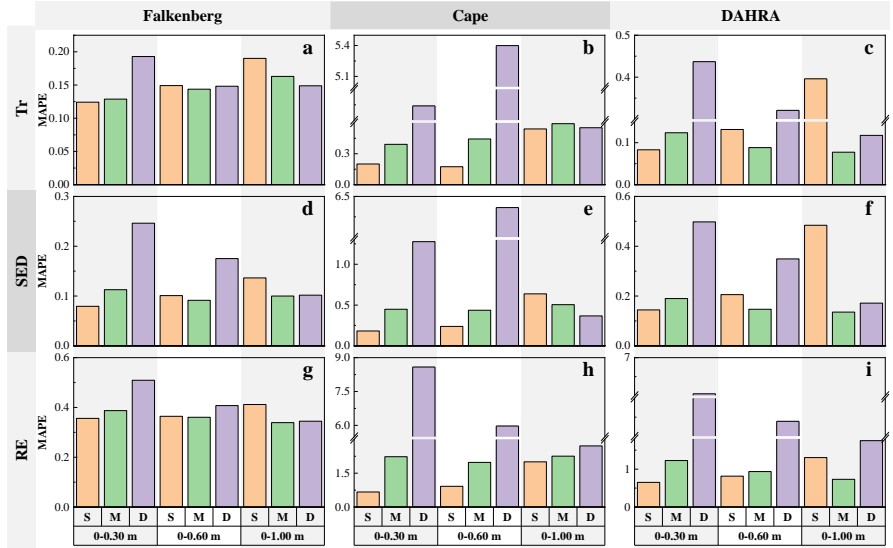

**Figure 6.** The MAPEs between expected and reference data-worth in the form of $T_r$, *SED*, and *RE* of potential soil

moisture observations in the surface (S), middle (M), and deep (D) layers, respectively, regarding the retrieval of

average soil moisture in the top 0.30 m, 0.60 m, and 1.00 m at three sites

**4.2 Effects of Observation Noise (TC1-1/TC4/TC5)**

Fig. 7 shows the probability distributions of the potential observation ensemble as well as their mean

and the corresponding actual observations of the surface SWC under different SWC noise levels.

Similarly, only the results on the 81st, 90th, and 99th days are displayed. It can be observed that a higher

noise level was not always detrimental but rather instead expanded the distribution width along the

SWC-axis and produced a flatter curve. The risk of failure of the generated realizations to "capture" the

real observations was thus reduced. Even on the 81st day at Falkenberg, for example, the increase in

SWC error variance from $0.01^2$ to $0.04^2$ facilitated a better agreement between the potential and actual

surface soil moisture, as revealed in Fig. 7 (a j s). Similar phenomena can also be found via a





comparison of Fig. 7e and Fig. 7n.

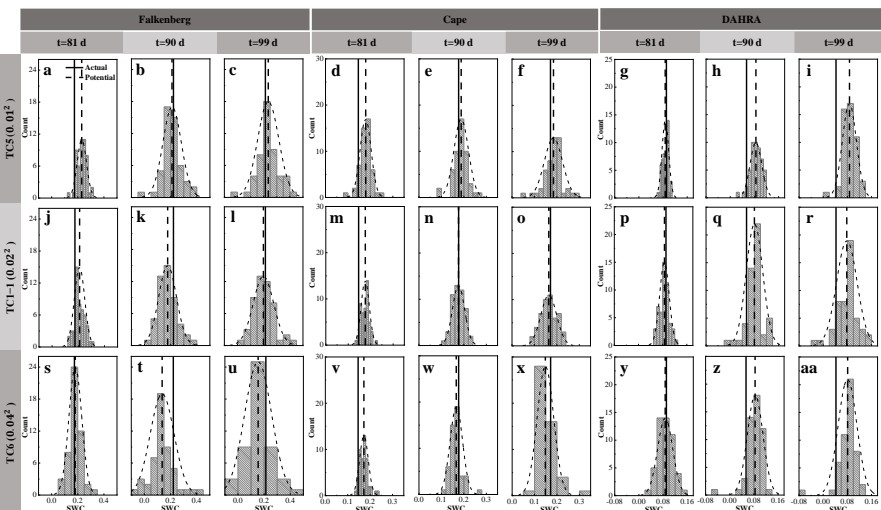

**Figure 7.** The probability distributions (dotted curved line) of potential observation realizations as well as their mean (dotted vertical line) and the corresponding actual soil water content (SWC) observations (solid line) in the surface layer on the 81st, 90th, and 99th day at three sites under different measurement error variances being $0.01^2$ (TC4), $0.02^2$ (TC1-1), and $0.04^2$ (TC5), respectively

Fig. 8 shows the temporal evolution and time-averaged MAPE of the expected and reference data-worth in the form of three information indices, respectively, under various noise levels. Some interesting findings can be obtained: (1) Overall, the potential SWC data corrupted by a lower noise level yielded larger data-worth with higher accuracy. (2) Nevertheless, the occurrence of rainfall events triggered a futile DW increase while also rendering the potential observations with appropriately magnified observation errors more valuable. For instance, a properly inflated observation error of $0.02^2$ on the 88th day at the Falkenberg site resulted in a notably higher data-worth than that of $0.01^2$, as highlighted by the dashed ellipse boxes in Fig. 8a and Fig. 8d. Furthermore, this increase in data-worth resulting from noise amplification was particularly evident in the form of $T_r$ over the other two metrics, as depicted in Fig. 8(a-c) and Fig. 8(d-i). At DAHRA, potential observations with an observation error of $0.02^2$ even produced a significantly higher $T_r$ value than that of $0.01^2$ throughout almost the entire simulation period (Fig. 8c). (3) As opposed to $T_r$ and $SED$ indices focusing only on the system uncertainty (variance or covariance), the expected $RE$, as a comprehensive mean-covariance-type





metric, was often more challenging to approach its reference counterparts with the largest MAPE at all

sites, as shown in Fig. 8(j-l).

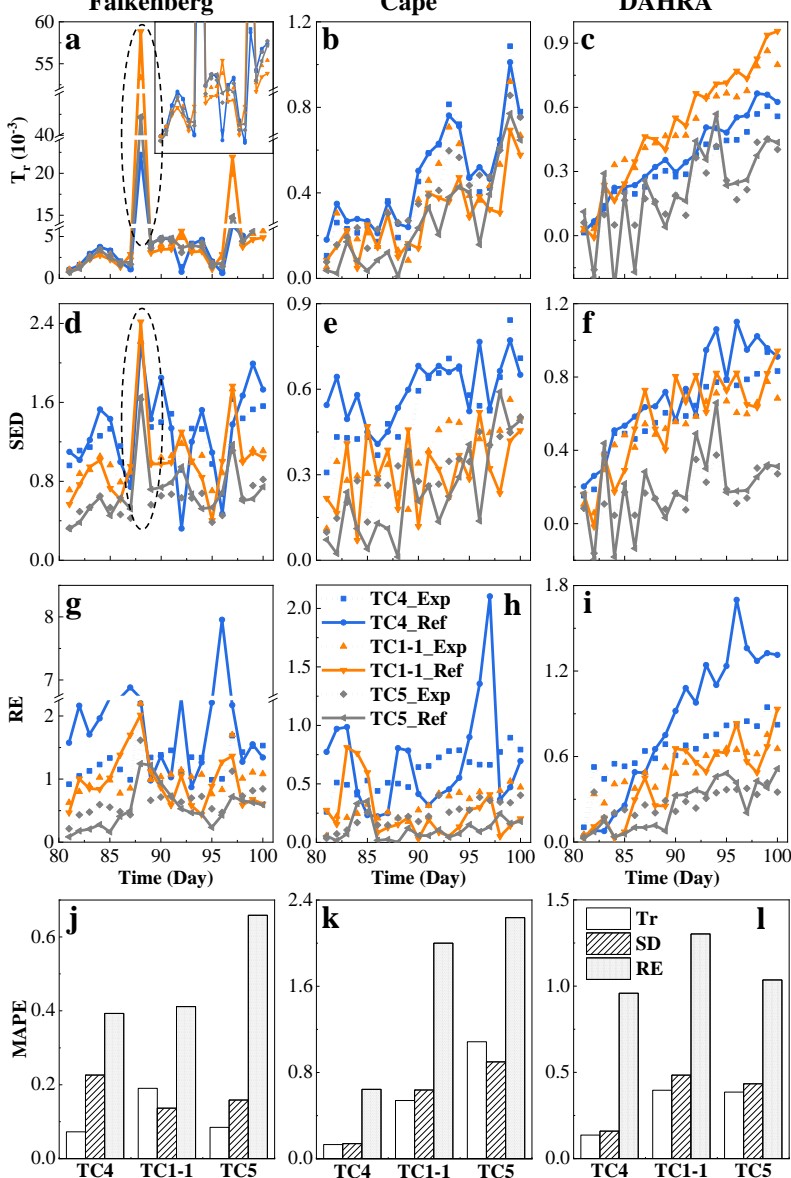

**Figure 8.** The temporal evolution (a-i) and time-averaged MAPEs (j-l) of the expected and reference data-worth in

the form of $T_r$, $SED$, and $RE$ at three sites, respectively, under different measurement error variances being $0.01^2$

(TC4), $0.02^2$ (TC1-1), and $0.04^2$ (TC5), respectively




### 4.3 Effects of Prior Data Content (TC1-1/TC6/TC7)

Fig. 9(a-i) depicts the temporal evolution of the expected and reference data-worth of the surface SWC in cases TC6, TC1-1, and TC7 with the 40-day, 80-day, and 180-day prior data content, respectively. Under normal circumstances, an increase in the available prior data content inevitably

entails a shrinkage in the DW of subsequent data due to the possibility of information redundancy. However, this seems to be valid only for a modest increase in prior data (from 40-day to 80-day) within our NP-DWA framework. The substantial augmentation in available data content from 80-day to 180-day instead resulted in a notably higher DW of the additional data (Fig. 9). Even more unexpectedly, this DW growth was prevalent across sites, regardless of the soil types and climatic

regimes. To clarify this anomaly, Fig. 10 further shows the predicted covariance matrixes of soil moisture profiles conditional on $\{A\}$ in prior stage and $\{A, B\}$ in preposterior stage in cases TC6, TC1-1, and TC7, respectively. Only the results from the $81^{st}$ day to $90^{th}$ day at Falkenberg are presented here. Our previous studies have demonstrated that although the mean values of potential samples can approach actual observations well in fully (Wang et al., 2021a) or partially (Zhang et al., 2019)

data-driven dynamical systems, their ensemble was apt to suffer from considerable uncertainty (Wang et al., 2021b). Unfortunately, augmented prior data, despite its potential to enrich available GP training scenarios, failed to prevent the non-convergence of $N_e = 50$ GP samples. In contrast, the additional noise associated with prior data supplementation could exacerbate the increase in the prior prediction uncertainty (i.e., $C_1$), as illustrated by a comparison between the first three columns of Fig. 10. It

should be highlighted that the fusion of $B$ enabled a notable reduction in the preposterior uncertainties (i.e., $C_2$) in the data assimilation system to a comparable level (the last three columns of Fig. 10), even with different prior data content. The gradual widening of the gap between $C_1$ and $C_2$ eventually yielded the highest data-worth with the maximum amount of prior data in test case TC7 (Fig. 9). This seems to alarm us that uncontrolled expansion of big data within fully data-driven systems may not be

beneficial. The adverse effects of extra noise may overshadow its original superiority in generalization capability. Access to high-quality and representative "small" data may constitute the key to the successful application of fully data-driven algorithms for reshaping soil moisture dynamics.

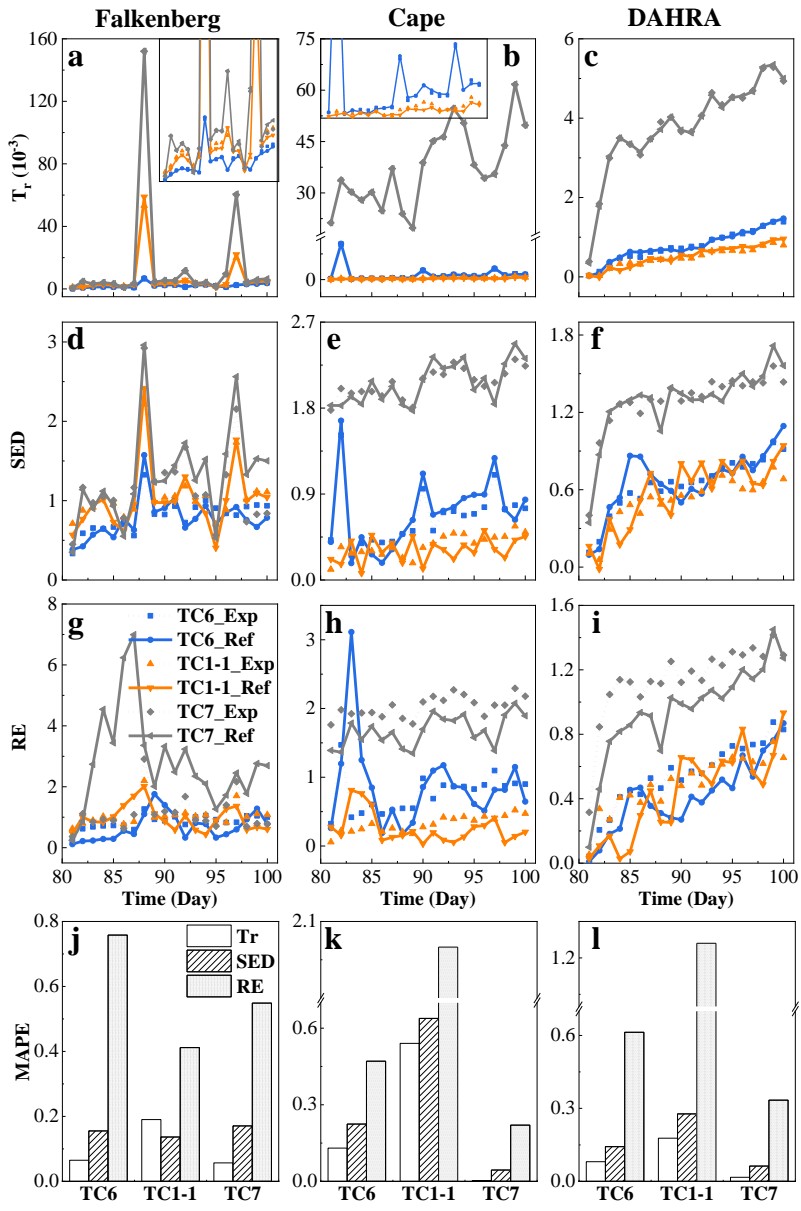

**Figure 9.** The temporal evolution (a-i) and time-averaged MAPEs (j-l) of the expected and reference data-worth in

the form of $T_r$, $SED$, and $RE$ at three sites for cases TC6, TC1-1, and TC7 with 40-day, 80-day, and 180-day prior

data content, respectively



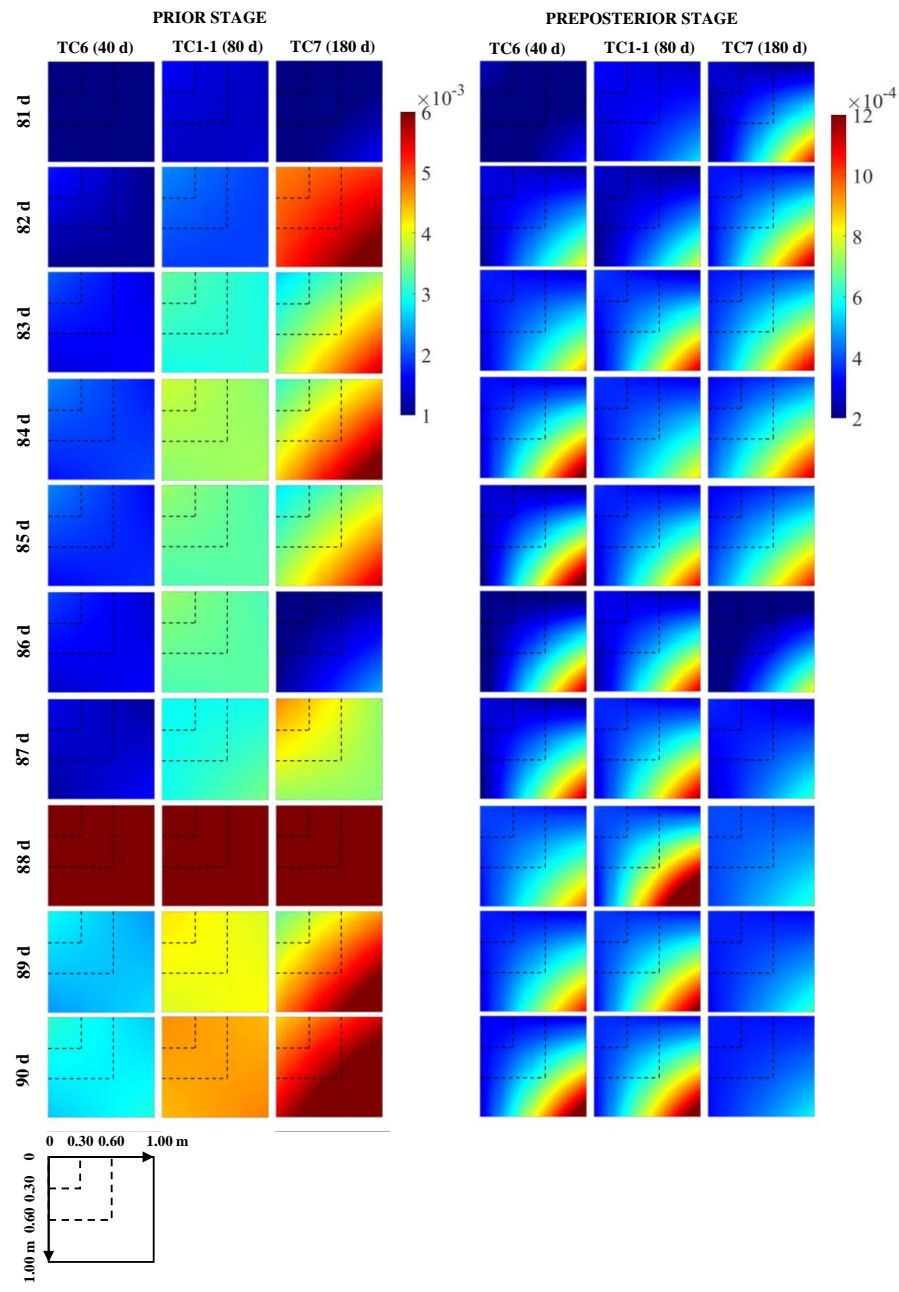

**Figure 10.** The covariance matrixes of soil moisture profiles from the 81st day to 90th day at Falkenberg in the prior

and preposterior stage for cases TC6, TC1-1, and TC7 with 40-day, 80-day, and 180-day prior data content,

respectively


Furthermore, Fig. 9(j-l) depicts the time-averaged MAPE in the expected and reference data-worth in cases TC6, TC1-1, and TC7, respectively. A comparison of Fig. 9(a-i) and Fig. 9(j-l) reveals some interesting findings: (1) similar to the results in Sect. 4.1, the potential measurements with the largest

expected (or reference) data-worth in TC7 are apt to possess the highest estimation accuracy of data-worth. (2) Local variations in data-worth at different sites respond slightly differently to the augmentation of prior data content. For instance, even with 180 days of available historical data, the DW spike induced by the unexperienced rainfall event on the 88[th] day at the Falkenberg has not been eliminated or diminished (Fig. 9(a d g)). However, similar DW surges on the 82[nd] day at Cape were

successfully mitigated as the amount of prior data content increased from 40-day (TC6) to 80-day (TC1-1) (Fig. 9(e h)). This is because the prior data at Falkenberg, even if augmented to 180-day, did not cover the rainfall event on the 88[th] day (Fig. 2b), whereas the 80-day training data at Cape already included the scenario on the 82[nd] (Fig. 2d). These results agree with the conclusions reported in Wang et al. (2020) that the diversity of scenarios in the training data is more decisive than the data volume

regarding the performance of data-driven methods. (3) Although inferior to $T_r$ and $SED$, the estimation accuracy of $RE$ is generally acceptable, especially when prior data is expanded to 180 days. This is certainly a remarkable improvement over the rather poor performance of $RE$ in traditional parametric data-worth analysis (Wang et al., 2018; Wang et al., 2020). This progress should be attributed to the radical abandonment of physical models in the NP-DWA, which prevented adverse

effects of the high nonlinearity of soil water flow in the propagation of uncertainties from input to output (i.e., soil moisture in this study). Direct mapping from regular meteorological data to SWC facilitated the identification of the soil moisture covariance matrix from potential observations.

### 4.4 Effects of potential observational combinations (TC1-1/TC8/TC9)

Fig. 11(a-i) compares the expected (and reference) data-worth of three combinations of potential

observations at different depths at the three sites. It can be seen that the composite data-worth of the alternative monitoring schemes exhibited an increasing pattern as the depth range of the observed SWC continues to expand downward. Nevertheless, the response of the different data-worth indicators and study sites to this vertical expansion of potential observations varied slightly. Further integration of $\theta_D$ in TC9 did not cause a marked increase in $T_r$ but yielded notably greater $SED$ and $RE$ values, especially

at the DAHRA site (Fig. 11(c f d)). This is undoubtedly due to the extra consideration of the latter two indicators for the non-diagonal elements of the covariance matrix or/and the behavior of the mean.





Moreover, the joint fusion of potential $\theta_S$ and $\theta_M$ failed to result in a sustained increase in $T_r$ and $RE$ at DAHRA, while creating a significant increase in composite DW at the other two sites. This could be attributed to the sandy soil texture at DAHRA (with the fraction of sand up to 90%, and $K_s$= 3.22 m/d),

resulting in the almost synchronous responses of the SWC at $z$=0.05 m and 0.50 m to the atmospheric boundary conditions (Fig. 2e) and thus triggering possible data redundancy.

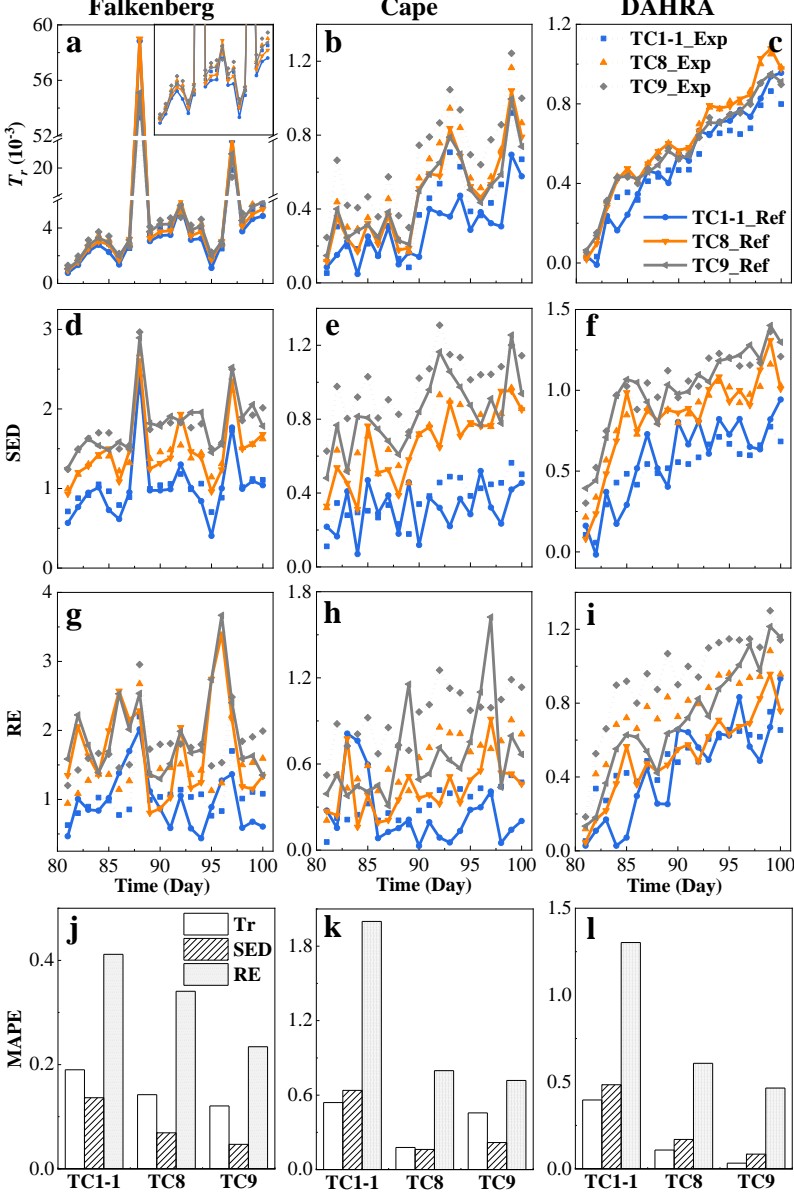

**Figure 11.** The temporal evolution (a-i) and time-averaged MAPEs (j-l) of the expected and reference data-worth



in the form of $T_r$, *SED*, and *RE* at three sites for cases TC1-1 (surface soil moisture), TC8 (surface & middle soil

moisture), and TC9 (surface & middle & deep soil moisture), respectively

Fig. 11(j-l) further shows the estimation accuracy of expected data-worth for the above three

potential observation combinations. Surprisingly, the increase in the number of potential observations,

while making it more difficult to "capture" actual SWC data, ends up significantly improving the

accuracy of the data-worth assessment. As shown in Fig. 11(j-l), as more potential observations along

the vertical direction were evaluated, the MAPEs between expected data-worth and its reference

counterparts decreased continuously. This phenomenon actually breaks the misconceptions about the

data-worth assessment accuracy in previous studies, i.e., that an excellent fit of potential observations is

equivalent to high-precision estimates of the corresponding data-worth. For the sake of explanation, Fig.

12 shows the predicted covariance matrixes of soil moisture profiles in cases TC1-1, TC8, and TC9

from the 81$^{st}$ day to 90$^{th}$ day at Falkenberg conditional on $\{\boldsymbol{A}\}$, $\{\boldsymbol{A}, \boldsymbol{B}\}$, and $\{\boldsymbol{A}, \boldsymbol{B}'\}$, respectively. It can

be found that compared to TC1-1, which only reduces the uncertainties in the surface SWC, the

integration of observations at multiple depths clearly reduces the uncertainties in the entire SWC

profiles to a considerably lower level. This ultimately facilitates better proximity between expected and

reference covariance matrixes, as revealed in the 4$^{th}$ and 7$^{th}$ columns in Fig. 12. The above results

suggest that the accuracy of data-worth assessment of potential observations does not only depend on

their capacity to "capture" actual measurements, but is also closely related to their correlation with the

variable of interest. We recall that similar phenomena also exist in the preceding test cases. For

example, the weaker correlation between surface SWC observations and $\theta_{1.00}^{ave}$ led to deterioration in

the DW estimation performance with the largest MAPE values (Fig.6(a d g)) even if the actual surface

observations could be suitably reproduced (Fig. 3(a-c)). Therefore, to enhance the reliability of

data-worth assessment, a strategy wherein potential observations at multiple depths were

simultaneously incorporated into existing DA systems was recommended in this study.





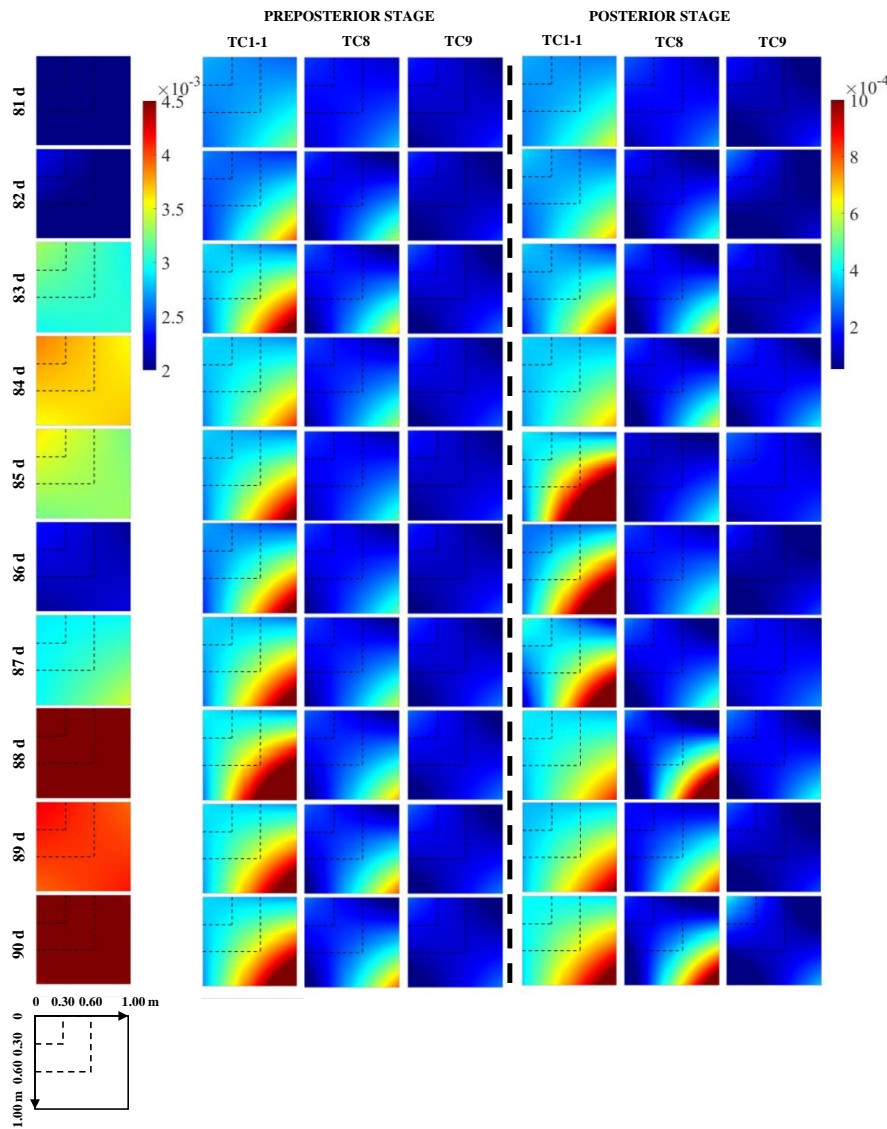

**Figure 12.** The covariance matrixes of soil moisture profiles from the 81st day to 90th day at Falkenberg before (column 1) and after potential (columns 2-4) and actual observations (columns 5-7) for cases TC1-1 (surface soil moisture), TC8 (surface & middle soil moisture), and TC9 (surface & middle & deep soil moisture) were fused, respectively

## 5 Conclusions

Conventional data-worth analysis for soil water problems depends on physical dynamic models.



Due to the widespread occurrence of model structural errors, it may lead to biased or wrong worth assessment. The strong nonlinearity of unsaturated flow further deteriorates the DW assessment performance in the retrieval of soil moisture profiles. This study proposed a nonparametric data-worth analysis method within a fully data-driven modeling framework. The information extracted from real-time soil moisture data after GP training and Kalman update was quantified with three representative types of indicators, i.e., variance- ($T_r$), covariance- ($SED$), and mean-covariance-type ($RE$) indicators. With the aid of a series of real-world cases, the ability and challenge of the NP-DWA in terms of the variables of interest, spatial location, observation error, and prior data content were assessed. The following conclusions were drawn:

(1) The proposed NP-DWA framework enabled an accurate assessment of the data-worth of potential observations regarding the reconstruction of purely data-driven soil water flow models prior to data collection. Nevertheless, the overall increasing trend of the DW from the sequential augmentation of additional observations was susceptible to interruptions by localized surges due to never-experienced atmospheric conditions within the NP-DWA framework. Fortunately, the adverse effects of anomalous GP extrapolation in our nonparametric approach could be suitably avoided by the enrichment of training scenarios in prior data. Moreover, the appropriate amplification of observational noise under extreme meteorological conditions also facilitated the alleviation of this biased estimates by enhancing the generalization capacity of dynamic models.

(2) The optimal observation depth shifted as the prediction target varied. In contrast to the notably higher DW of surface SWC observations within the conventional DW analysis framework, middle SWC observations tended to exhibit considerably higher robustness in the construction of model-free soil moisture dynamic models. This should be attributed to the ability of the SWC in the middle layer to effectively reduce the predictive uncertainty of the entire soil moisture profiles due to its optimal representativeness. The inherent delayed response of soil moisture profiles to rainfall events allowed this advantage of middle SWC prevalent across sites, even becoming increasingly pronounced with increasing delay effect.

(3) Although the addition of prior data content could greatly improve the estimation accuracy of the expected DW, the ensuing observation noise could substantially increase the uncertainty in a purely data-driven DA system, leading to potentially higher data-worth of subsequent observations. Hence, high-quality and representative small data may be regarded as a better choice than unfiltered big data.

(4) An alternative monitoring strategy with a larger data-worth was prone to a higher DW assessment accuracy within the proposed NP-DWA framework. Specifically, the performance of data-worth assessment was jointly determined by '3Cs', i.e., capacity of potential observation realizations to "capture" actual observations, correlation of potential observations with the predicted variables of interest, and choice of DW quantitative indicators. Furthermore, the direct mapping from regular meteorological data to SWC in our nonparametric method facilitated the identification of the soil moisture covariance matrix (especially the cross-covariance) due to its alleviation of highly nonlinearity of soil water flow problems. Hence, satisfactory estimation accuracy could also be achieved even with covariance-related data-worth metrics (i.e., the *SED* and *RE*).

## ACKNOWLEDGEMENTS

This work was supported by the National Natural Science Foundation of China Grants U2243235 and 51979200, and the Open Research Fund of Guangxi Key Laboratory of Water Engineering Materials and Structures Grant GXHRI-WEMS-2020-06.

## CODE/DATA AVAILABILITY

The code/data that support the findings of this study are available from the corresponding author upon reasonable request.

## AUTHOR CONTRIBUTION

**Yakun Wang**: Conceptualization, Methodology, Software, Writing–original draft. **Xiaolong Hu**: Conceptualization, Software. **Lijun Wang**: Methodology. **Jinmin Li**: Data curation, Methodology. **Lin Lin**: Supervision. **Kai Huang**: Data curation. **Liangsheng Shi**: Writing – review & editing, Supervision.

## COMPETING INTERESTS

The contact author has declared that none of the authors has any competing interests.

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
