# Peer review of "Data worth analysis within a model-free data assimilation framework for soil moisture flow"

_Hydrology and Earth System Sciences, 2023_

## Author Comment (AC1)

**Reply on RC1**

General comments: This study proposes a comprehensive data-driven framework for selecting the optimal observing operations (data-worth analysis) and updating the predictions for soil moisture dynamics. The fully data-driven approach provides a complement to physics-based models, especially for complex real-world scenarios. While the quality of the manuscript is good, there are still some issues that require clarification.

Specific comments:

1. A major concern is the conclusions drawn from applying the Gaussian processes and EnKF assimilation techniques. While efficient and simple to implement, these methods have inherent limitations such as excessively smooth predictions (GP) and optimality only for Gaussian linear problems (EnKF). As the soil moisture dynamics are not fully met by these assumptions, the proposed method may experience difficulties, such as the mentioned localized surges. Therefore, some conclusions "high-quality and small data may be better than unfiltered big data" and "the soil water content in the middle layer exhibits remarkable superiority in comparison to the surface with its highest-level variability" may be case-specific rather than generalizable. It is important to consider other data-driven and assimilation methods, such as deep neural networks, particle filtering, and MCMC, leading to potentially different outcomes. I would like to see some clarifications regarding this issue.

**Answer:**

Thank you for your constructive comments. We have accepted your suggestions and evaluated a new NP-DWA framework where EnKF is replaced by particle filtering (PF). **Fig. S1** depicts the expected data-worth of potential observations of $\theta_S$, $\theta_M$, and $\theta_D$ regarding the retrieval of $\theta_{0.30}^{ave}$, $\theta_{0.60}^{ave}$, and $\theta_{1.00}^{ave}$, respectively. A comparison of **Fig. S1** and **Fig. 4** reveals that the spatio-temporal changes of expected data-worth under these two assimilation methods are remarkably similar. This demonstrates the generalizability of our proposed framework and related conclusions under different data assimilation schemes. To avoid duplication of research, we are sorry that we finally decided not to add the results of PF in the main text, but rather to include them as supplementary material in the revised manuscript (please see Lines 165-170 and Supplement).

In addition, we also tested two other NP-DWA frameworks where GP was replaced by support vector machines (SVM) and random forests (RF), respectively. The temporal changes of expected data-worth metrics are depicted in **Fig. S2**. Only the results at DAHRA are presented here. A comparison of **Fig. S2** and **Fig. 4** indicates that although the magnitude and trends of data-worth vary slightly across different machine learning methods, the selection of the optimal monitoring depths for specific targets is quite consistent. For example, the optimal observation depth shifted as the prediction target varied, and soil water content in the middle layer robustly exhibited remarkable superiority in the construction of model-free soil moisture models. Moreover, the performance comparison of various machine learning algorithms in reproducing soil moisture dynamics has been widely discussed in previous studies (Dubois et al., 2021; Liu et al., 2020; Prakash et al., 2018). In particular, the ability of GP to reproduce the nonlinearity of soil water problems has also been demonstrated in (He et al., 2023; Ju et al., 2018; Wang et al., 2021). Therefore, we finally decided only to include these results as supplementary material as well in the revised manuscript.

[Figure]

**Figure S1.** The expected data-worth of potential soil moisture observations in the surface, middle, and deep layers in the form of trace ($T_r$), Shannon entropy difference (*SED*), and relative entropy (*RE*), respectively, regarding the retrieval of average soil moisture in the top 0.30 m, 0.60 m, and 1.00 m at three sites, when EnKF is replaced by particle filtering (PF) in the proposed NP-DWA framework

[Figure]

**Figure S2.** The expected data-worth of potential soil moisture observations in the surface, middle, and deep layers in the form of $T_r$, *SED*, and *RE* regarding the retrieval of average soil moisture in the top 0.30 m, 0.60 m, and 1.00 m at DAHRA site, when GP is replaced by support vector machine (SVM) and random forest (RF) in the proposed NP-DWA framework, respectively

**References:**

Dubois, A., Teytaud, F. and Verel, S., 2021. Short term soil moisture forecasts for potato crop farming: A machine learning approach. Computers and Electronics in Agriculture, 180: 105902.

He, L. et al., 2023. Physics-constrained Gaussian process regression for soil moisture dynamics. Journal of Hydrology, 616: 128779.

Ju, L., Zhang, J., Meng, L., Wu, L. and Zeng, L., 2018. An adaptive Gaussian process-based iterative ensemble smoother for data assimilation. Advances in water resources, 115: 125-135.

Liu, Y., Jing, W., Wang, Q. and Xia, X., 2020. Generating high-resolution daily soil moisture by using spatial downscaling techniques: A comparison of six machine learning algorithms. Advances in Water Resources, 141: 103601.

Prakash, S., Sharma, A. and Sahu, S.S., 2018. Soil moisture prediction using machine earning. IEEE, pp. 1-6.

Wang, Y. et al., 2021. A nonparametric sequential data assimilation scheme for soil moisture flow. Journal of Hydrology, 593: 125865.

2. It is recommended that the methodology section of this paper be better presented. Specifically, the problem setup for moisture prediction, an explicit list of the contents of vectors X and y should be provided prior to section 2.1. This will enable the reader to better understand the proposed data-driven framework.

**Answer:**

Thank you for your valuable suggestions. We have revised the methodology section. A clearer description of vectors $X$ and $y$ has also been added in section 2.1 of the revised manuscript (please see Lines155-160 and 170-185).

3. Some techniques have been proposed for better performance in nonlinear problems, e.g., restart, iterations. How will these techniques perform in NP-DWA?

**Answer:**

We thank the reviewer for the constructive comments. In fact, the procedure of constructing GP models in a sequential manner in our NP-DWA framework resembles a restart operation. At any time step $t=k$, the construction of the GP model does not solely rely on the information from the previous time step, instead, its training data includes all available soil moisture data from $t=1$ to $t=(k-1)$. This restart-like operation ensures that the training database is sequentially augmented to include more diverse training scenarios, so that actual observations can be accurately "captured" by the generated potential observation samples. Ultimately, the accuracy (or reliability) of our NP-DWA framework for data-worth assessment can be guaranteed. Related descriptions have been added in the revised manuscript (please see Lines 170-185). The performance improvements of these techniques such as restart and iterations for our NP-DWA will be explored in our future study.

4. L31:"An alternative monitoring strategy with a larger data-worth was prone to a higher DW assessment accuracy within the proposed NP-DWA framework" This sentence is meaningless and should be removed.

**Answer:**

We have accepted the reviewer's suggestion and deleted this sentence (please see Lines 35 and 665).

5. Please provide the dimensionality for all the involved vectors and matrices.

**Answer:**

We have added the dimensionality for all the involved vectors and matrices in Section 2 of the revised manuscript.

---

## Author Comment (AC3)

**Reply on RC2**

The manuscript by Wang et al. presents a framework for determining data worth of soil moisture measurements. The framework uses Gaussian process regression (GP) to replace unsaturated flow models; GP is combined with EnKF to evaluate the prior, post-posterior, and posterior (regarding potential new data) distributions of variables of interest (i.e., soil moisture averaged for varying portions of the soil column). The change of distributions was summarized using three indices to determine data worth. The framework was demonstrated using three soil columns from ISMN using several test cases to illuminate the roles of prior data length, observation noise, and combinations of potential new data.

Overall the manuscript is clearly organized and results are thoroughly described and discussed. Data worth analysis in a model-free framework (using machine learning) is novel. I therefore recommend it to be published in HESS after minor revisions. Below are detailed comments, most of which are intended to improve clarity and generalizability of the presented framework.

Some discussion is needed to support the conjunctive use of GP and EnKF. EnKF is very commonly used with deterministic models such as Hydrus for data assimilation and propagation of uncertainty in time. GP is capable of assimilating newly available data by simply training the model again once new data is available and calculating the mean and covariance of a variable of interest. Given this, it seems that the data-worth framework can be done using GP alone, without EnKF. Some discussion is needed to help readers understand the framework design. For example, what is the role of EnKF in improving accuracy of data worth estimation or enabling the framework to be used for machine learning algorithms other than GP? Will EnKF result in covariance matrices different from those calculated by GP?

**Answer:**

We thank the reviewer for the constructive suggestions. In fact, the necessity of the conjunctive use of GP and EnKF has been discussed in detail in our previous studies (Wang et al., 2021; Wang et al., 2021). As stated in (Wang et al., 2021), on the one hand, the fusion of EnKF can effectively reduce the risk of unreasonable spatio-temporal interpolation in GP models, ultimately enhancing the robustness of such purely data-driven models; On the other hand, by combining with Kalman update, the forecast cross-covariance ($C_k^f H^T$) between the state ($Y_k^f$) and the predictions corresponding to available observations ($HY_k^f$) in Eqs (6-7) constrained the otherwise high error covariances of state variables at unobserved depths, which resulted in a significantly reduced uncertainty for this hybrid method relative to GP alone. To keep this manuscript more focused, we finally decided to only add a brief explanation of the conjunctive use of GP and EnKF (please see Lines 160-165), without adding extra cases with GP alone in the revised manuscript.

**References:**

Wang, Y. et al., 2021. A nonparametric sequential data assimilation scheme for soil moisture flow. Journal of Hydrology, 593: 125865.

Wang, Y., Shi, L., Zhang, Q. and Qiao, H., 2021. A gradient-enhanced sequential nonparametric data assimilation framework for soil moisture flow. Journal of Hydrology, 603: 126857.

It is unclear from my reading (1) what depth(s) are used as prior data for training GP, (2) what

specific depth(s) are considered for potential data (theta_s, m, d), and (3) do the depths for prior data and potential new data overlap?

**Answer:**

We are sorry for the confusion caused by our unclear description. (1) Prior data for training GP includes the soil water content at all observed depths during the prior stage (from $t=1$ to $t=T_p$), i.e., $z$=0.08, 0.15, 0.30, 0.45, 0.60, and 0.90 m at Falkenberg, $z$=0.05, 0.10, 0.20, 0.50, and 1.00 m at Cape, and $z$=0.05, 0.10, 0.50, and 1.00 m at the DAHRA. (2) The depth of the potential soil moisture data is different in different test cases. For example, as listed in **Table 2**, the potential data in TC1-1, TC2-1, TC3-1, TC4, TC5, TC6, and TC7 refers to soil moisture in the surface layer ($\theta_S$), i.e., $z$=0.08 m at Falkenberg and $z$=0.05 m at Cape and DAHRA. (3) The depths of the prior and potential new data in this study partially overlapped due to the limited depths of observations under real-world circumstances. For example, TC1-1 at Falkenberg used soil moisture observations taken at six depths ($z$=0.08, 0.15, 0.30, 0.45, 0.60, and 0.90 m) over the first 80 d as prior data, and the generated soil moisture at $z$=0.08 m over the last 20 d as potential data. We have added the relevant descriptions in the revised manuscript (please see Lines 340-355).

Line 305 - how is the noise level used in the computations? For GP or for KF? Is the noise level specified or estimated when training GP?

**Answer:**

Thank you for your carefully reading. Noises from soil moisture observations are considered in both GP and EnKF in this study. At any time step $t=k$ during GP modelling, the observed time series from $t=1$ to $(k-1)$ are corrupted by the prescribed observation noises satisfying Gaussian distribution to obtain $N$ sets of training data. Subsequently $N$ sets of GP models are constructed independently, to generate $\boldsymbol{Y}_k^f = \left[\boldsymbol{y}_{k,1}^f, \boldsymbol{y}_{k,2}^f, ..., \boldsymbol{y}_{k,m}^f, ..., \boldsymbol{y}_{k,N}^f\right]^T$ in Eq. 7 (please see Lines 175). In the analysis stage of EnKF, the real-time observation $\boldsymbol{d}_k^{obs}$ perturbed by the specified noise was assimilated via Eq. 8 (please see Lines 230-235). Considering the difficulty of determining the observation noise under real-world circumstances, the noise level is artificially specified in this study. We have added the relevant explanations in the revised manuscript (please see Lines 315 and 369).

Line 375 - I don't have a specific comment here, but would like to highlight that the difference in data worth between a physical model and a machine learning model is very interesting and a key contribution of this study.

**Answer:**

Thank you for your valuable recognition. Considering that the data-worth analysis in physical models has been discussed in detail in our previous study (Wang et al., 2018), this study did not add the corresponding test cases, but directly compared the findings in (Wang et al., 2018) with the results of the proposed NP-DWA. As you mentioned, the comparison in data-worth between physical and machine learning models can help modelers better understand the impact of the ways data being utilized on its worth. We have accepted your suggestions and further highlight this difference in the revised manuscript (please see Lines 20-30, 420-425, 565-575, and 640-650).

**References:**

Wang, Y. et al., 2018. Sequential data-worth analysis coupled with ensemble Kalman filter for soil

water flow: A real-world case study. Journal of Hydrology, 564: 76-88.

Predictions (variables of interest) considered in this study are depth-averaged soil moisture. It would be of more interest to the broader hydrologic community to discuss the potential application of the presented framework for other types of predictions, e.g., ET, infiltration.

**Answer:**

Thank you for your constructive suggestions. This study used the GP model to reconstruct the nonlinear relationship between multiple variables (including time, depth, precipitation, and air temperature) and soil moisture. Therefore, the expected data-worth of future monitoring programs regarding the estimation of depth-averaged soil moisture can be evaluated. Our future study will further discuss the application of the NP-DWA framework for other types of predictions, e.g., ET, infiltration.

Line 365: "matrixes" should be "matrices"

**Answer:**

Thank you for your carefully reading. We have revised "matrixes" to "matrices".

---

## Editor Decision (ED1)

[revised manuscript text omitted]
[x_1^1, x_1^2, \dots, x_1^{q_1}, \dots, x_p^1, x_p^2, \dots, x_p^{q_p}, \dots, x_{k-1}^1, x_{k-1}^2, \dots, x_{k-1}^{q_{k-1}}\right]^T \tag{1}$$

$$\boldsymbol{y}_{1:(k-1)} = \left[y_1^1, y_1^2, \dots, y_1^{q_1}, \dots, y_p^1, y_p^2, \dots, y_p^{q_p}, \dots, y_{k-1}^1, y_{k-1}^2, \dots, y_{k-1}^{q_{k-1}}\right]^T \tag{2}$$

where $\boldsymbol{X}_{1:(k-1)}$ and $\boldsymbol{y}_{1:(k-1)}$ denote a collection of all available $x$ and $y$ from $t$=1 to $(k$-1), respectively; $q_p$ denotes the number of available observations at $t$=$p$ ($p$=1, 2, …, $k$-1). In this paper we assume that the number of available observations at each time step is identical, i.e., $q_1 = q_2 = \dots = q_{k-1} = q$. Hence the dimensions of matrix $\boldsymbol{X}_{k-1}$ and vector $\boldsymbol{y}_{k-1}$ are $q(k-1) \times d$ and $q(k-1)$, respectively.

 As defined in Rasmussen (2003) and Williams and Rasmussen (2006), a GP  model can be fully specified by a mean function $\mu(\boldsymbol{x})$ and covariance function $k(\boldsymbol{x}, \boldsymbol{x}')$, i.e., $G(\boldsymbol{x}) \sim N(\mu(\boldsymbol{x}), k(\boldsymbol{x}, \boldsymbol{x}'))$. In this study, a linear mean function and an anisotropic squared exponent covariance function are specified (Zhang et al., 2019) as:

$$\mu(\boldsymbol{x}) = \boldsymbol{\beta}^{\mathrm{T}} \boldsymbol{x} \tag{3}$$

$$k(\boldsymbol{x}, \boldsymbol{x}') = \sigma^2 exp\left[-\sum_{l=1}^{d} \frac{(x_l - x_l')^2}{\tau_l^2}\right] \tag{4}$$

where $\boldsymbol{\beta}$ is vector containing $d$ linear coefficients, i.e., $\boldsymbol{\beta} = \{\beta_1, \beta_2, \dots, \beta_d\}$;  $\sigma^2$ controls the marginal variance in the output; and $\boldsymbol{\tau} = \{\tau_1, \tau_2, \dots, \tau_d\}$  determines the dependence strength in each of the component directions of $\boldsymbol{x}$.

 Then, the hyperparameters of the $GP_k$ , $\boldsymbol{\phi} = \{\boldsymbol{\beta}, \sigma^2, \boldsymbol{\tau}\}$, can be inferred from the training datasets $\{\boldsymbol{X}_{1:(k-1)}$$, \boldsymbol{y}_{1:(k-1)}$$\}$ via log marginal likelihood maximization:

$$L = \log p(\boldsymbol{y}_{1:(k-1)}|X_{1:(k-1)}, \boldsymbol{\phi}) = -\frac{1}{2}(y - \mu)^{\mathrm{T}}\Sigma^{-1}(y - \mu) - \frac{1}{2}\log|\Sigma| - \frac{n}{2}\log 2\pi$$
$$= -\frac{1}{2}(\boldsymbol{y}_{1:(k-1)} - \boldsymbol{\mu})^{\mathrm{T}}\Sigma^{-1}(\boldsymbol{y}_{1:(k-1)} - \boldsymbol{\mu}) - \frac{1}{2}\log|\Sigma| - \frac{n}{2}\log 2\pi \tag{5}$$

where $\boldsymbol{\mu}$ denotes the prior mean  vector with the dimension of $q(k-1)$, and $\boldsymbol{\Sigma}$ denotes the covariance matrix whose elements in the $i$th row and $j$th column constitute $\Sigma_{ij} = k(\boldsymbol{x}_i, \boldsymbol{x}_j)$ ($i$=1, 2, …, $q(k-1)$; $j$=1, 2, …, $q(k-1)$). The GPML MATLAB toolbox (version 4.2), as documented in Williams and Rasmussen (2006), was adopted for GP inference in this study

(http://www.gaussianprocess.org/gpml/code/matlab/doc/).

In this study, the entire soil moisture profile at $t=k$ is expected to be forecasted. Assuming that the total number of nodes of the vertical one-dimensional soil profile is $N_n$, then the input at the current time step is $XX_k^* = [x_k^1, x_k^2, \ldots, x_k^{N_n}]^T$ with the dimension of $N_n \times d$. The corresponding output vector $y_k^f$ with the dimension of $N_n$ can be calculated as:

$$y_k^f \bcancel{y^*} = \mu^* + \Sigma^{*T} \Sigma^{-1} (y_{1:(k-1)} \bcancel{y} - \mu) \tag{46}$$

where $\mu^*$ denotes the prior mean of $\mu(XX_k^* X^*)$ with the dimension of $N_n$ and $\Sigma^*$ is calculated as $\Sigma_i^* = k(x_k^i x_i, x_j X_j^*)$ ($i$=1, 2, …, $N_n$; $j$=1, 2, …, $q(k-1)$).

As a collection of $y_k^f$ from $N$ GP models, the resultant forecasted state vector $Y_k^f$ at $t=k$ can be represented as:

$$Y_k^f = [y_{k,1}^f, y_{k,2}^f, \ldots, y_{k,m}^f, \ldots, y_{k,N}^f]^T \tag{7}$$

where $y_{k,m}^f$ denotes the forecasted state vector of interest for $GP_k^m$ ($m$=1, 2, …, $N$); the dimension of $Y_k^f$ is $N \times N_n$; superscripts $f$ denotes forecast.

**2.2 The Kalman Update in Nonparametric Data Assimilation Scheme**

~~Similar to the conventional EnKF method (Evensen, 2003), the model free DA strategy also comprises forecast and analysis steps. At the forecast step at $t=k$, $N_1$ GP dynamic models are constructed in the light of $\{(X_{1:k-1}, y_{1:k-1})_1, \ldots, (X_{1:k-1}, y_{1:k-1})_i, \ldots, (X_{1:k-1}, y_{1:k-1})_{N_1}\}$ independently via Eqs. 1–3. Here, $(X_{1:k-1}, y_{1:k-1})$ represents all available data points from $t=1$ to $(k-1)$ and $i$ is the ensemble member index. Hence an ensemble of $y^*$ at the current time step, $Y_k^f$, can be forecasted via Eq. 4, which can be expressed as:~~

$$\bcancel{Y_k^f = [y_{k,1}^*, y_{k,2}^*, \ldots, y_{k,i}^*, \ldots, y_{k,N_1}^*]^T} \tag{5}$$

In the analysis step of the EnKF, for any ensemble member $m$ at $t=k$, the state vector can be updated by combing GP model predictions and observations $d_k^{obs}$:

$$\cancel{Y} y^a_{k,m} = \cancel{Y} y^f_{k,m} + K_k(d^{obs}_{k,m} - H\cancel{Y} y^f_{k,m}) \tag{68}$$

where $y\cancel{Y}^a_{k,m}$ denotes the improved estimates for realization $m$ by conditioning on the observed information at $t=k$; $H$ is the observation operator with the dimension of $q \times N_n$, which represents the relationship between the state and observation vectors; superscript $a$ indicates analysis. and $d^{obs}_{k,m}$ with the dimension of $q$ denotes the observation vector at $t=k$ for the $m^{\text{th}}$ ensemble member of $d^{obs}_k$. It should be emphasized that the relationship between observations at $t=k$, $d^{obs}_k$ and their true values $y_k = [y^1_k, y^2_k, \ldots, y^q_k]^T$ can be expressed as follows,

$$d^{obs}_k = y_k + \varepsilon_k \tag{9}$$

where $\varepsilon_k$ with the dimension of $N \times q$ represents measurement error vector which is assumed to be zero-mean Gaussian with $R_k$; $R_k$ denotes the error covariance matrix of the observations with the dimension of $q \times q$.

The Kalman gain at $t=k$, $K_k$, with the dimension of $N_n \times q$ can be defined as:

$$K_k = C^f_k H^T (H C^f_k H^T + R_k)^{-1} \tag{710}$$

where  $C^f_k$ with the dimension of $N_n \times N_n$ is the covariance matrix of the state vector at $t=k$, which can be approximated as:

$$C^f_k \approx \frac{1}{\cancel{N_e}N - 1} \sum_{\cancel{i}m=1}^{\cancel{N_e}N} \left\{ [y^f_{k,\cancel{i}m} - \langle Y^f_k \rangle][y^f_{k,\cancel{i}m} - \langle Y^f_k \rangle]^T \right\} \tag{811}$$

where $\langle Y^f_k \rangle$ denotes the ensemble mean of $Y^f_k$.

[revised manuscript text omitted]

---

## Author Response (AR2)

Dear Editor,

Thank you for your letter dated Jun 22, 2023. On behalf of my co-authors, we appreciate you very much for giving us an opportunity to revise our manuscript (HESS-2023-34) entitled "Data worth analysis within a model-free data assimilation framework for soil moisture flow". We are also grateful for your constructive comments on our manuscript. We have carefully taken your suggestions into consideration in preparing our corrections, which are all valuable and very helpful for improving our work. In addition, we have also updated the reference list according to the technical requirements of the HESS.

We hope that the revised manuscript is more suitable for the publication in "Hydrology and Earth System Sciences".

Yours sincerely,

All correspondence regarding the manuscript should be addressed to:

Yakun Wang

Key Laboratory of Agricultural Soil and Water Engineering of in Arid and Semiarid Areas

Northwest A & F University

Yangling, Shaanxi, 712100, China

Should you have any questions, please let me know (E-mail: wangyakun@nwafu.edu.cn). Your consideration of the manuscript is greatly appreciated.

Line 158: "observation time, depth" Please specify what you measured at this time and depth. Why is the measured quantity not included in $x$? Or are these the times and depths for which the soil moisture content is sought?

**Answer:**

Thank you for your careful reading. These are indeed the times and depths for which the soil moisture content is sought. We have improved the relevant descriptions in the revised manuscript (please see Lines 154-156).

Lines 169-171: scheme; Supplementary

**Answer:**

We have modified these words in the revised manuscript (please see Lines 165-168).

Line 335: "2cm grids" Space missing.

**Answer:**

We have corrected this error in the revised manuscript (please see Line 312).

Table 2: "$0.02^2$" Is this a footnote? If so, where is it? If it is not, why is this there?

**Answer:**

Thank you for your careful reading. In fact, $0.02^2$ indicates that the observed error variance of soil moisture is 0.02 squared. In order to avoid misunderstanding, we have modified $0.02^2$ to 0.0004 in the revised manuscript (please see Table 2).

Line 570: "whose resultant deterioration" What does this mean here? Do you mean the following? ...unresolved model structural errors. When these lead to a deteriorating DW assessment performance, this cannot be compensated by assimilating more prior data

**Answer:**

We are sorry for your confusion due to our inappropriate expressions. As you stated, this means that "…unresolved model structural errors. When these lead to a deteriorating DW assessment performance, this cannot be compensated by assimilating more prior data". We have improved these descriptions in the revised manuscript (please see Lines 544-546).

Line 572: "On the contrary" In contrast,

**Answer:**

   We have modified these words in the revised manuscript (please see Line 547).

Line 630: I stand by my first assessment aht this is all a bit qualitative, which is a bit unexpected in a paper striving to express data worth in quantitative terms.

That being said, I do not consider this a major stumbling block, and the reviewers did not bring this up, so I will not hold up the paper because of this.

**Answer:**

   Thanks for your constructive comments. In the future, we will focus more on the quantitative research on data worth.